# A review of current knowledge concerning PM$_{2.5}$ chemical composition, aerosol optical properties, and their relationships across China

Jun Tao[1], Leiming Zhang[2,*], Junji Cao[3], Renjian Zhang[4]

[1]South China Institute of Environmental Sciences, Ministry of Environmental Protection, Guangzhou, China

[2]Air Quality Research Division, Science and Technology Branch, Environment and Climate Change Canada, Toronto, Canada

[3]Key Laboratory of Aerosol Chemistry and Physics, Institute of Earth Environment, Chinese Academy of Sciences, Xi'an, China

[4]Key Laboratory of Regional Climate-Environment for Temperate East Asia, Institute of Atmospheric Physics, Chinese Academy of Sciences, Beijing, China

*Corresponds to: leiming.zhang@canada.ca

**Abstract**
To obtain a thorough knowledge of $PM_{2.5}$ chemical composition and its impact
on aerosol optical properties across China, existing field studies conducted after the
year 2000 are reviewed and summarized in terms of geographical, inter-annual, and
seasonal distributions. Annual $PM_{2.5}$ was up to six times of the National Ambient Air
Quality Standards (NAAQS) in some megacities in northern China. Annual $PM_{2.5}$
was higher in northern than southern cities, and higher in inland than coastal cities.
In a few cities with data longer than a decade, $PM_{2.5}$ showed a slight decrease only in
the second half of the past decade, while carbonaceous aerosols decreased, sulfate
($SO_4^{2-}$) and ammonium ($NH_4^+$) remained at high levels, and nitrate ($NO_3^-$) increased.
The highest seasonal averages of $PM_{2.5}$ and its major chemical components were
typically observed in the cold seasons. Annual average contributions of secondary
inorganic aerosols to $PM_{2.5}$ ranged from 25% to 48%, and those of carbonaceous
aerosols ranged from 23% to 47%, both with higher contributions in southern
regions due to the frequent dust events in northern China. Source apportionment
analysis identified secondary inorganic aerosols, coal combustion, and traffic
emission as the top three source factors contributing to $PM_{2.5}$ mass in most Chinese
cities, and the sum of these three source factors explained 44% to 82% of $PM_{2.5}$
mass on annual average across China. Biomass emission in most cities, industrial
emission in industrial cities, dust emission in northern cities, and ship emission in
coastal cities are other major source factors, each of which contributed 7-27% to
$PM_{2.5}$ mass in applicable cities.
The geographical pattern of scattering coefficient ($b_{sp}$) was similar to that of
$PM_{2.5}$, and that of aerosol absorption coefficient ($b_{ap}$) was determined by elemental
carbon (EC) mass concentration and its coating. $b_{sp}$ in ambient condition of
RH=80% can be amplified by about 1.8 times of that under dry condition. Secondary
inorganic aerosols accounted for about 60% of aerosol extinction coefficient ($b_{ext}$) at
relative humidity (RH) greater than 70%. The mass scattering efficiency (MSE) of
$PM_{2.5}$ ranged from 3.0 to 5.0 $m^2$ $g^{-1}$ for aerosols produced from anthropogenic
emissions and from 0.7 to 1.0 $m^2$ $g^{-1}$ for natural dust aerosols. The mass absorption
efficiency (MAE) of EC ranged from 6.5 to 12.4 $m^2$ $g^{-1}$ in urban environments, but
the MAE of water-soluble organic carbon was only 0.05 to 0.11 $m^2$ $g^{-1}$. Historical
emission control policies in China and their effectiveness were discussed based on
available chemically resolved $PM_{2.5}$ data, which provides the much needed
knowledge for guiding future studies and emissions policies.

# Contents

## 1. Introduction

Knowledge of spatiotemporal variations of chemical and optical properties of atmospheric aerosols is needed in addressing regional and global air quality and climate issues (Fuzzi et al., 2015; Ginoux et al., 2012; Li et al., 2016c; Liao et al., 2015; Monks et al., 2009; Qian et al., 2015). Aerosol concentrations across China have been at extremely high levels in the recent two decades, largely caused by rapidly increased energy consumption (Chan and Yao, 2008; Fang et al., 2009; Guan et al., 2014; Wang and Hao, 2012; Zhang et al., 2013b). The frequency of haze has also been increased significantly due to light extinction of atmospheric aerosols, especially $PM_{2.5}$ (Li and Zhang, 2014; Pui et al., 2014; Watson, 2002). The Ministry of Environmental Protection of China thus promulgated NAAQS to include daily and annual $PM_{2.5}$ standards starting in early 2012. As a result, real-time $PM_{2.5}$ data in 74 Chinese cities have been recorded since 2013.

Light extinction, the sum of light scattering and absorption, is controlled by not only $PM_{2.5}$ levels, but also its chemical composition, size-distribution and hygroscopic potential of its major components, and meteorological conditions (Hand and Malm, 2007a; Malm et al., 2003; Pitchford et al., 2007; Zhang et al., 2014a). High humidity combined with large fractions of hygroscopic chemical components (e.g. sulfate, nitrate, ammonium, and some organic matter) can enhance light extinction and haze intensity (Liu et al., 2011; Liu et al., 2013b; Zhang et al., 2015b; Zieger et al., 2013). A large number of studies has been conducted in China in recent years investigating $PM_{2.5}$ composition, aerosol optical properties, aerosol

hygroscopic properties, and haze formation mechanisms (Guo et al., 2014; Jing et al.,
2015; Liang et al., 2016; Liu et al., 2011; Liu et al., 2012; Pan et al., 2009; Tao et al.,
2014b; Wang et al., 2015b; Yan et al., 2008; Yan et al., 2009; Yang et al., 2011b;
Zheng et al., 2016). However, knowledge of long-term trends of $PM_{2.5}$ concentration,
especially its major chemical components, is still limited (Fontes, 2017), and few
studies have focused on the geographical pattern of $PM_{2.5}$ composition across China
and its impact on aerosol optical properties (Li et al., 2017a). The present study aims
to gain such knowledge through a thorough review of available studies.
Considering the large number of publications, only ground measurement data of
chemical composition of $PM_{2.5}$, aerosol scattering and absorption coefficients, and
aerosol hygroscopic properties published after the year 2000 in scientific papers of
Science Citation Index (SCI) journals are reviewed and summarized in this study. A
total of about 150 articles met the above criteria including 100 articles on $PM_{2.5}$
chemical composition and source apportionment, 40 articles on aerosol optical
properties, and 10 articles on aerosol hygroscopic properties. Many of these articles
focused on several of the biggest cities such as, Beijing, Shanghai, Guangzhou and
Hong Kong, while other studies focused on cities including Tianjin, Shijiazhuang,
Jinan, Nanjing, Hangzhou, Fuzhou, Xiamen, Shenzhen, Chengdu, Chongqing, Xi'an,
Lanzhou, Zhengzhou, Wuhan, Changsha, Haikou and several background sites (Fig.
2). Geographical and temporal patterns of $PM_{2.5}$ and its major chemical components
including $SO_4^{2-}$, $NO_3^-$, $NH_4^+$, organic carbon (OC), and EC, and aerosol optical
properties are generated, source-apportionment analysis results are summarized, and
relationships between aerosol optical properties and PM$_{2.5}$ chemical composition are
explored. Recommendations are also provided for alleviating PM$_{2.5}$ levels and
reducing haze occurrence.
**2. Spatiotemporal patterns of PM$_{2.5}$ and its major chemical**
**components**
In this section, available measurements of chemically resolved PM$_{2.5}$ are
reviewed and summarized in terms of geographical distributions, inter-annual
variations, and seasonal patterns. Measurements are grouped based on geographical
regions, such as the Beijing-Tianjin-Hebei (BTH) in North China Plain, the Yangtze
River Delta (YRD), the Pearl River Delta (PRD), the Sichuan Basin, and other regions
(Fig. 1). Five dominant chemical components of PM$_{2.5}$ (SO$_4^{2-}$, NO$_3^-$, NH$_4^+$, OC, and
EC) are discussed in detail. Data reviewed in this section are all listed in Table S1 of
the supplement document.
**2.1 PM$_{2.5}$ mass**
Filter-based measurements of PM$_{2.5}$ were mainly carried out in urban cities of
BTH (Beijing, Tianjin, Shijiazhuang, and Chengde), YRD (Shanghai, Nanjing, and
Hangzhou), PRD (Guangzhou, Hong Kong, Zhongshan, and Shenzhen), Sichuan
basin (Chongqing, Chengdu, and Neijiang), and other cities (e.g., Jinan, Xi'an,
Lanzhou, Zhengzhou, Wuhan, Changsha, Fuzhou, Xiamen, and Haikou).
Geographical characteristics of annual PM$_{2.5}$ are first discussed followed by
interannual variations and seasonal patterns.
**2.1.1 Geographical distributions**
Annual mean $PM_{2.5}$ mass concentrations in major cities in different regions are
plotted in Fig. 2a. Regional annual mean and standard deviation (SD) values were
calculated using annual mean data of all the cities where data are available. Regional
annual mean $PM_{2.5}$ was 115±29, 96±28, 50±16, and 100±35 µg m$^{-3}$ in BTH (Chen et
al., 2014c; Duan et al., 2006; He et al., 2001; He et al., 2012; Song et al., 2006a; Tian
et al., 2016; Wang et al., 2005; Yang et al., 2011a; Yang et al., 2011b; Zhang et al.,
2013a; Zhao et al., 2013c; Zhou et al., 2015a; Zíková et al., 2016), YRD (Feng et al.,
2009; Li et al., 2015a; Li et al., 2016a; Liu et al., 2015; Ming et al., 2017; Wang et al.,
2006; Wang et al., 2016a; Ye et al., 2003; Zhao et al., 2015b), PRD (Hagler et al.,
2006; Huang et al., 2013; Louie et al., 2005a; Tao et al., 2014c; Tao et al., 2017), and
Sichuan basin (Chen et al., 2014d; Tao et al., 2013a; Tao et al., 2014a; Wang et al.,
2017a; Yang et al., 2011b), respectively, which was 3 to 6 times, 2 to 3 times, 1 to 2
times, and 3 to 6 times of NAAQS, respectively.
Within each region, the highest annual average $PM_{2.5}$ concentration was
observed in Shijiazhuang (191 µg m$^{-3}$), Hangzhou (121 µg m$^{-3}$), Guangzhou (65 µg
m$^{-3}$) and Chengdu (111 µg m$^{-3}$) in BTH, YRD, PRD and Sichuan basin, respectively.
Outside the above-mentioned four regions, annual $PM_{2.5}$ at individual cities was
183±25 µg m$^{-3}$ (Geng et al., 2013; Wang et al., 2015a; Wang et al., 2017b), 177±15 µg
m$^{-3}$ (Shen et al., 2009; Wang et al., 2015c; Zhang et al., 2011b), 89 µg m$^{-3}$ (Wang et al.,
2016b), 149 µg m$^{-3}$ (Yang et al., 2012), 110±4 µg m$^{-3}$ (Xiong et al., 2017; Zhang et al.,
2015a), 106 µg m$^{-3}$ (Tang et al., 2017), 66±22 µg m$^{-3}$ (Zhang et al., 2011a; Zhang et al.,
2012a; Zhang et al., 2016), 44 µg m$^{-3}$ (Xu et al., 2012c) and 21 µg m$^{-3}$ (Liu et al.,
2017a) in Zhengzhou, Xi'an, Lanzhou, Jinan, Wuhan, Changsha, Xiamen, Fuzhou and
Haikou, respectively. These PM$_{2.5}$ levels were comparable to some of the cities within
the four regions, e.g., Zhengzhou, Xi'an and Jinan to Shijiazhuang, Wuhan to Nanjing
and Chengdu, and Fuzhou and Xiamen of Fujian province to Guangzhou. Cities in Fig.
2a are rearranged in Fig. 2b from north to south latitudes. Except for a few cities, such
as Chengde and Beijing, there was a decreasing trend in annual PM$_{2.5}$ mass
concentration with decreasing latitude. Moreover, annual PM$_{2.5}$ mass concentrations
in western or inland cities were higher than those in eastern or coastal cities along the
same latitudes. The geographical patterns of the filter based PM$_{2.5}$ measurements
agreed well with the online monitoring of PM$_{2.5}$ in 31 provincial capital cities in
China (Wang et al., 2014b).

Filter-based measurements of PM$_{2.5}$ at rural sites in China were limited and

mainly conducted at Shangdianzi of Beijing, Conghua and Tianhu of Guangzhou, and
Hok Tsui of Hong Kong (Hagler et al., 2006; Lai et al., 2016; Louie et al., 2005a;
Zhao et al., 2013c). Rural PM$_{2.5}$ was around half of that in the cities of the same
region. A similar geographical pattern was seen in rural PM$_{2.5}$ as in the urban, e.g.,
annual PM$_{2.5}$ at the rural site of BTH (Shangdianzi) was 72 µg m$^{-3}$, which was 2 times
of that (35 µg m$^{-3}$) at the rural sites of PRD.
**2.1.2 Inter-annual variations**

Data collected in most cities were within a three-year time window, except in

Beijing, Shanghai and Guangzhou where PM$_{2.5}$ data spanned for more than a decade
(1999-2014) (Fig. 3). Inter-annual variations in $PM_{2.5}$ in Beijing were small, ranging
from 100 to 128 µg m$^{-3}$, similar to the trends in the online data, which ranged from
65 to 83 µg m$^{-3}$ during 2004-2012 (Liu et al., 2014b). The lower concentrations of
the online than filter $PM_{2.5}$ data is likely caused by volatilization loss of nitrate and
organic matters from the tapered element oscillating microbalances (TEOM), which
operated at 50℃ during the online sampling. These results suggested that there was
no evidence that $PM_{2.5}$ pollution has been significantly improved in Beijing during
the 15 year study period despite the many control measures that have been
implemented. The impact of local pollution controls in Beijing has likely been offset
by regional pollutant transport (Li et al., 2015b). In Shanghai, $PM_{2.5}$ in 2003-2006
(94 µg m$^{-3}$) (Feng et al., 2009; Wang et al., 2006) and 2009 (94 µg m$^{-3}$) (Zhao et al.,
2015b) was nearly 50% higher than earlier years (e.g., 65 µg m$^{-3}$ in 1999-2000) (Ye
et al., 2003); although it decreased slightly to 58 µg m$^{-3}$ in 2011-2013 (Wang et al.,
2016a; Zhao et al., 2015b), it increased rapidly back to 95 µg m$^{-3}$ in 2013-2014
(Ming et al., 2017). In Guangzhou, $PM_{2.5}$ in 2002-2003 (71 µg m$^{-3}$) (Hagler et al.,
2006) and in 2009-2010 (77 µg m$^{-3}$) (Tao et al., 2014c) were kept at stable levels and
then decreased to 48 µg m$^{-3}$ in 2014 (Tao et al., 2017).
**2.1.3 Seasonal patterns**

In BTH, the highest seasonal average $PM_{2.5}$ concentrations were observed in

winter and the lowest in summer in all the cities with seasonal variations up to
factors of 1.7, 1.5, 1.6 and 1.8 in Beijing (Cao et al., 2012b; Chan et al., 2005; Chen
et al., 2014a; Dan et al., 2004; Duan et al., 2006; Han et al.,2014; He et al., 2001;
Huang et al., 2014b; Lin et al., 2016; Okuda et al., 2011; Pathak et al., 2011; Song et
al., 2006a; Song et al., 2007; Sun et al., 2004; Sun et al., 2006; Tan et al., 2016a; Tao
et al., 2016a; Tao et al., 2015a; Tian et al., 2015; Wang et al., 2005; Yang et al.,
2005a; Yang et al., 2016; Zhang et al., 2013a; Zhao et al., 2013c), Tianjin (Cao et al.,
2012b; Gu et al., 2010; Gu et al., 2011; Li et al., 2009; Tian et al., 2016; Zhao et al.,
2013c), Shijiazhuang (Zhao et al., 2013c), and Chengde (Zhao et al., 2013c),
respectively. It is noted that major pollutant sources in BTH were located south of
Hebei province and the prevailing winds in BTH were from the north in winter and
from the south in summer (Li et al., 2016b; Lu et al., 2010; Lu et al., 2011; Wang et
al., 2013; Xu et al., 2011). The location and distribution of major industrial sources,
intensity of local minor sources such as winter heating, and prevailing wind
directions together caused the slightly different magnitudes of seasonal variations
among the four cities discussed above. Moreover, extreme weather events such as
weakening monsoon circulation, depression of strong cold air activities, strong
temperature inversion, and descending air masses in the planetary boundary layer
also played important roles in the strong $PM_{2.5}$ pollution during winter (Niu et al.,
2010; Wang et al., 2014c; Zhao et al., 2013d). Several extreme wintertime air
pollution events in recent years covered vast areas of northern China and were all
correlated to some extent with extreme weather conditions (Zou et al., 2017).

In YRD, the highest seasonal average $PM_{2.5}$ concentrations were also observed

in winter and the lowest in summer with seasonal variations up to factors of 2.3, 1.9
and 2.0 in Nanjing (Li et al., 2015a; Li et al., 2016a; Shen et al., 2014; Yang et al.,
2005b), Shanghai (Cao et al., 2012b; Cao et al., 2013; Feng et al., 2009; Feng et al.,
2012a; Hou et al., 2011; Huang et al., 2014a; Huang et al., 2014b; Ming et al., 2017;
Pathak et al., 2011; Wang et al., 2006; Wang et al., 2016a; Ye et al., 2003; Zhao et al.,
2015b), and Hangzhou (Cao et al., 2012b; Liu et al., 2015), respectively. In PRD,
most urban site $PM_{2.5}$ studies were also accompanied with rural site studies (Andreae
et al., 2008; Cao et al., 2004; Cao et al., 2012b; Cui et al., 2015; Duan et al., 2007; Fu
et al., 2014; Ho et al., 2006a; Huang et al., 2007; Huang et al., 2014b; Jahn et al.,
2013; Jung et al., 2009; Lai et al., 2007; Lai et al., 2016; Liu et al., 2014a; Louie et al.,
2005a; Tan et al., 2009; Tan et al., 2016c; Tao et al., 2009; Tao et al., 2014c; Tao et al.,
2015b; Tao et al., 2017; Wang et al., 2012; Yang et al., 2011b). Although the highest
seasonal average $PM_{2.5}$ was also observed in winter, the season with the lowest
concentration was not consistent between the sites, e.g., in summer in Guangzhou and
in spring in Hong Kong. This was likely caused by warm/hot temperatures in this
region and frequent precipitation in warm seasons, and thus small differences between
spring and summer, e.g., $PM_{2.5}$ concentration of 32 µg m$^{-3}$ in summer in Guangzhou
(Cao et al., 2004; Cao et al., 2012b; Duan et al., 2007; Ho et al., 2006a; Lai et al.,
2007; Louie et al., 2005a) and 29 µg m$^{-3}$ in spring in Hong Kong (Louie et al., 2005a).
Seasonal variations were up to a factor of 1.9 in both cities. $PM_{2.5}$ at rural sites in
PRD generally doubled during dry seasons (autumn and winter) compared to wet
seasons (spring and summer) due to frequent precipitation scavenging of aerosols in
wet seasons (Cheung et al., 2005; Dai et al., 2013; Fu et al., 2014; Griffith et al., 2015;
Hu et al., 2008; Lai et al., 2016).
Similar seasonal patterns as above were also observed in cities of other regions
in China, such as Chengdu (Tao et al., 2013a; Tao et al., 2014a), Zhengzhou (Geng et
al., 2013), Jinan (Yang et al., 2012) and Fuzhou (Xu et al., 2012b), with seasonal
variations between a factor of 1.8 and 2.5. In conclusion, the highest seasonal average
$PM_{2.5}$ observed in winter at all urban sites in China was likely due to high emissions
from winter heating and/or poor pollutant dispersion.
## 2.2 Major chemical components of $PM_{2.5}$
It is well known that OC, EC, $SO_4^{2-}$, $NO_3^-$ and $NH_4^+$ are the dominant chemical
components in $PM_{2.5}$. Thus, only studies having synchronous measurements of $PM_{2.5}$
and the above-mentioned five major components were discussed below. Note that for
most cities only short-term measurements were available, however, for Beijing,
Shanghai and Guangzhou, existing studies span a period of 15 years (2000-2014).
To ensure the comparability of the data collected using different instruments,
measurement uncertainties were first briefly discussed here. Most studies in China
analyzed OC and EC using DRI carbon analyzer or Sunset carbon analyzer.
IMPROVE is the most widely used thermal/optical protocol for OC and EC analysis
for DRI analyzer while NIOSH is the one for the Sunset analyzer. OC and EC
measured by the two analyzers are comparable if the same analysis protocol is used.
For example, Wu et al. (2011) showed that OC from the Sunset analyzer was only 8%
lower than that from the DRI analyzer, while EC was only 5% higher. However, when
different protocols were used by the two analyzers, the differences were much larger,
e.g., EC from NIOSH was almost 50% lower than that from IMPROVE (Chow et al.,
2010; Yang et al., 2011a). Note that OC and EC were also measured using a CHN
elemental analyzer in 2001-2002 in Beijing, which uses a similar protocol to NIOSH
(Duan et al., 2006). In any case, the measurement uncertainties of total carbon (TC,
the sum of OC and EC) were less than 10% (Chow et al., 2010; Wu et al., 2011).
The ions including $SO_4^{2-}$, $NO_3^-$ and $NH_4^+$ were measured by ion chromatography.
Measurement uncertainties should be less than 15% in most cases under strict QA/QC
procedures (Orsini et al., 2003; Trebs et al., 2004; Weber et al., 2003), but could be
larger for ammonium nitrate ($NH_4NO_3$) since it can evaporate from the filters before
chemical analysis under high temperature and low RH conditions, and this applies to
both quartz fiber filter and Teflon filter (Keck and Wittmaack, 2005; Weber et al.,
2003). The loss of $NO_3^-$ due to evaporation was found to range from 4% to 84%
depending on the ambient temperature (Chow et al., 2005). Although the exact
magnitudes of measurement uncertainties cannot be determined for $NO_3^-$ and $NH_4^+$,
they are not expected to affect significantly the inter-annual variations discussed
below for the three cities (Beijing, Shanghai, and Guangzhou) considering the small
year-to-year temperature changes.
**2.2.1 The Beijing-Tianjin-Hebei region**
**2.2.1.1 Inter-annual variations in Beijing**
Chemically-resolved PM$_{2.5}$ data in BTH covering multiple-years are only
available in Beijing and the inter-annual variations are discussed for this city below
(Duan et al., 2006; He et al., 2001; Song et al., 2006a; Yang et al., 2011b; Zhang et
al., 2013a; Zhao et al., 2013c). Inter-annual variations of OC and EC were generally
small, e.g., a factor of 1.5 for OC and 1.8 for EC (Fig. 3a). OC decreased from
23.6-25.8 µg m$^{-3}$ in earlier years (1999-2006) to below 17.6 µg m$^{-3}$ after 2008. EC
increased from 6.3 µg m$^{-3}$ in 1999-2000 to 9.9 µg m$^{-3}$ in 2001-2002, and then
gradually decreased to 5.7 µg m$^{-3}$ in 2009-2010. TC increased from 29.8 µg m$^{-3}$ in
1999-2000 to 32.7-35.7 µg m$^{-3}$ in 2001-2006, and then decreased to 23.3 µg m$^{-3}$ in
2009-2010. The nearly 30% reduction in TC in recent years in Beijing can be taken
as a real trend since measurement uncertainties were believed to be around 10% as
mentioned above. OC is produced from both primary emissions and secondary
formation, while EC (also known as black carbon or BC) is mainly from primary
emissions. The anthropogenic emission for OC and BC over the entire China showed
an increasing trend in 1996-2010 (Lu et al., 2011), while BC emissions showed a
slightly decreasing trend in Beijing and Tianjin in 2005-2009 (Qin and Xie, 2012).
Meanwhile, BC emissions sharply increased in Hebei province in 2005-2009. The
amount of BC emissions in Hebei province was much higher than the sum of those
in Beijing and Tianjin (Qin and Xie, 2012). Thus, the decrease of EC concentration
in Beijing was likely dominated by local emission reduction instead of regional
transport from Hebei province.
Annual $SO_4^{2-}$ concentration increased slightly during 1999-2010 and ranged
from 10.2 µg m$^{-3}$ to 16.4 µg m$^{-3}$ in Beijing. $SO_2$ emission in China increased by
about 60% during 2000-2006 and then decreased about 9% during 2006-2010 due to
the compulsory flue-gas desulfurization equipment applied in power plants (Lu et al.,
2011). However, the sum of the $SO_2$ emissions in BTH (including Beijing, Tianjin,
and Hebei province) increased sharply from 2097 Gg year$^{-1}$ in 2000 to 2916 Gg
year$^{-1}$ in 2004, and further slightly increased to 2998 Gg year$^{-1}$ in 2007 before
sharply decreased to 1821 Gg year$^{-1}$ in 2010 (Lu et al., 2010; Zhao et al., 2013a). A
continued increase in $SO_2$ emission was found in Hebei province, which accounted
for more than 50% of the total $SO_2$ emission in BTH. In contrast, $SO_2$ emissions in
Beijing continued decreasing. Surface annual $SO_2$ concentrations in Beijing
gradually decreased from 56 µg m$^{-3}$ to 35 µg m$^{-3}$ during 2006-2009
(http://www.zhb.gov.cn/). Thus, the persistently high concentrations of $SO_4^{2-}$ in
Beijing was largely due to regional transport from Hebei province, noting that the
lifetime of $SO_4^{2-}$ is longer than that of $SO_2$.
$NO_3^-$ concentrations were relatively steady (7.4-10.9 µg m$^{-3}$) during 1999-2006,
but sharply increased to 15.9 µg m$^{-3}$ in 2009-2010 in Beijing. Both $NO_x$ ($NO_2$+NO)
emissions and satellite $NO_2$ vertical column densities synchronously increased
during 2000-2010 in China (Zhang et al., 2012b; Zhao et al., 2013b). Different from
those of $SO_2$ emissions, $NO_x$ emissions in all the cities and provinces in BTH
showed increasing trends in 2005-2010. $NO_x$ emissions in Beijing slightly increased
from 410 Gg year$^{-1}$ in 2005 to 480 Gg year$^{-1}$ in 2010 (Zhao et al., 2013b). However,
annual average surface $NO_2$ concentrations in Beijing showed a decreasing trend and
fluctuated in the range of 49 - 66 µg m$^{-3}$ during 2006-2009 (http://www.zhb.gov.cn/).
There were some inconsistences between the trends of surface $NO_2$ concentrations
and column $NO_2$ or $NO_x$ emissions, likely due to the impact of photochemical
reactions on surface $NO_2$ concentrations in urban areas. To some extent, the
increasing trend of $NO_3^-$ in Beijing was likely related to the increases in $NO_x$
emissions in both Beijing and the surrounding cities or provinces.
Considering the potential large uncertainties in $NH_4^+$ measurements, its trends
should only be discussed qualitatively. $NH_4^+$ concentrations were relatively steady in
Beijing during 1999-2006, ranging from 5.7 to 7.3 $\mu g \ m^{-3}$. $NH_3$ emissions changed
little (13400-13600 Gg year$^{-1}$) before 2005 in China, and increased slightly in BTH
region during 2003-2010 (Zhou et al., 2015b). The small increase of $NH_4^+$ in
2009-2010 in Beijing was consistent with the $NH_3$ emission trend in this region
(Zhang et al., 2013a; Zhao et al., 2013c). Moreover, the increase of $NO_3^-$ in Beijing
was also an important factor contributing to the increase of $NH_4^+$.
In summary, a decreasing trend was identified in TC and increasing trends were
found for $SO_4^{2-}$, $NO_3^-$ and $NH_4^+$ in Beijing. The inter-annual variations in EC agreed
with the local emission trends in Beijing, but those in $SO_4^{2-}$, $NO_3^-$ and $NH_4^+$
agreed more with the regional scale emission trends of their respective gaseous
precursors in BTH rather than the local emission trends in Beijing. Nonlinear
responses of concentration changes of these aerosol components to their respective
emission trends were found, demonstrating other important factors potentially
affecting aerosol formation. It is worth to note that several recent studies have
highlighted the important role $NO_2$ might play in sulfate formation in the polluted
environment in China (Cheng et al., 2016; Wang et al., 2016c; Xie et al., 2015a).
Nevertheless, the aqueous $SO_2 + H_2O_2/O_3$ oxidation should still be the dominant
mechanism in most cases, especially at a background site (Lin et al., 2017). The
aqueous $SO_2$ + oxygen (catalyzed by Fe (III)) reaction can also be important under
heavy haze conditions in north China (Li et al., 2017b). Extensive measurements of
stable oxygen are needed to confirm the relative contributions of different sulfate
formation mechanisms.
**2.2.1.2 Relative contributions to PM$_{2.5}$**
To investigate the relative contributions of dominant chemical components to
PM$_{2.5}$ mass, the measured PM$_{2.5}$ mass was reconstructed based on $SO_4^{2-}$, $NO_3^-$,
$NH_4^+$, OM (organic matter), and EC. The conversion factor between OC and OM
was 1.8 considering the prevailing biomass burning in BTH (Cheng et al., 2013a; Du
et al., 2014a).
Data collected in 2009-2010 were first discussed since multiple cities in BTH
have data during this period   (Zhang et al., 2013a; Zhao et al., 2013c). Secondary
inorganic aerosols (the sum of sulfate, nitrate and ammonium) contributed 36-39%
of PM$_{2.5}$ annually in the majority of the cities, but only 25% in Chengde, a tourist
city located in the northeast part of BTH and 200 kilometers away from Beijing.
Generally, the percentage contribution of secondary inorganic aerosols to PM$_{2.5}$
decreased with decreasing PM$_{2.5}$ level, e.g., from Shijiazhuang to Tianjin, Beijing,
and then Chengde, a phenomenon that is consistent with what was found within the
same city but for different pollution levels in a winter season (Tao et al., 2015a).
Carbonaceous aerosols contributed 29-32% to PM$_{2.5}$ in most cities, but as high as
45% in Chengde, and had an opposite trend to that of secondary inorganic aerosols in
terms of city-to-city variations. At the rural site Shangdianzi near Beijing, secondary
inorganic aerosols and carbonaceous aerosols accounted for 42% and 32%,
respectively, of the $PM_{2.5}$ mass, which were not significantly different from those in
cities located south of Yanshan Mountain. The sum of secondary inorganic aerosols
and carbonaceous aerosols accounted for 65%-70% of the $PM_{2.5}$ mass in cities of
BTH.

In Beijing where data are available for more than a decade, secondary inorganic

aerosols accounted for 28% of $PM_{2.5}$ on average and ranged from 23% to 31% from
year to year. Carbonaceous aerosols accounted for 43% of $PM_{2.5}$ and ranged from
29% to 55%. Seasonal average contributions of secondary inorganic aerosols were
generally higher in warm seasons than in cold seasons in most cities, and an opposite
trend was found for carbonaceous aerosols (Fig. 4). For example, secondary
inorganic aerosols contributed 32%, 41%, 28% and 32% in spring, summer, autumn
and winter, respectively, to $PM_{2.5}$ in Beijing, while carbonaceous aerosols
contributed 35%, 30%, 44% and 45% (Cao et al., 2012b; Duan et al., 2006; He et al.,
2001; Huang et al., 2014b; Lin et al., 2016; Pathak et al., 2011; Song et al., 2006a;
Song et al., 2007; Sun et al., 2004; Tao et al., 2015a; Tian et al., 2015; Zhang et al.,
2013a; Zhao et al., 2013c). A similar seasonal trend was also observed in other BTH
cities, e.g., secondary inorganic aerosols accounted for 42-53% of $PM_{2.5}$ mass in
summer and only 15-35% in winter, while carbonaceous aerosols accounted for
16-34% in summer and 42- 60% in winter in Tianjin (Cao et al., 2012b; Gu et al.,
2011; Li et al., 2009; Zhao et al., 2013c), Shijiazhuang (Zhao et al., 2013c) and
Chengde (Zhao et al., 2013c). Higher carbonaceous aerosols in winter should be
related to heating activities and biomass burning in this region (Cheng et al., 2013a;
Duan et al., 2004; Tao et al., 2016b; Wang et al., 2007; Yang et al., 2016).
**2.2.2 The Yangtze River Delta region**
**2.2.2.1 Inter-annual variations in Shanghai**
Chemically-resolved $PM_{2.5}$ data in YRD covering multiple-years are only
available in Shanghai (Ming et al., 2017; Wang et al., 2016a; Ye et al., 2003; Zhao et
al., 2015b). Inter-annual variations of OC in this city were within a factor of 1.6 for
OC and a factor of 4.1 for EC (Fig. 3b). OC concentrations were relatively steady
(14.0-14.9 µg $m^{-3}$) during 1999-2009, but sharply decreased to 9.9-10.1 µg $m^{-3}$ in
2011-2014. EC varied in the range of 4.1 to 6.5 µg $m^{-3}$ during 1999-2009, and also
sharply decreased to 1.6-2.1 µg $m^{-3}$ in 2011-2014. TC decreased from 19.5 µg $m^{-3}$
during 1999-2009 to 11.9 µg $m^{-3}$ in 2011-2014, or nearly 40% reduction, which was
much higher than the known measurement uncertainties. Noticeable reduction of OC
and EC occurred after 2010 Shanghai World Expo, which resulted in a decrease of
TC after 2010. BC emission slightly decreased in Shanghai in 2005-2009, but
increased in the adjacent Zhejiang and Jiangsu provinces (Qin and Xie, 2012). BC
emissions in Jiangsu province were much higher than the sum of those in Shanghai
and Zhejiang. Thus, the decreased EC concentration in Shanghai was largely a result
of local emission reductions.
Annual $SO_4^{2-}$ concentration decreased from 14.0 µg $m^{-3}$ in 1999-2000 to the
range of 11.7 µg $m^{-3}$ to 12.5 µg $m^{-3}$ during 2009-2014. The trend of $SO_2$ emission in
YRD generally agreed with that across the entire China, which showed an increasing
trend during 2000-2006 and a decrease one during 2006-2010 (Lu et al., 2011). The
annual variations in $SO_2$ emission in YRD (including Shanghai, Jiangsu, and
Zhejiang) were relative small, ranging from 3171 Gg year$^{-1}$ in 2000, 3506 Gg year$^{-1}$
in 2004, 3376 Gg year$^{-1}$ in 2007, and to 3397 Gg year$^{-1}$ in 2010 (Lu et al., 2010;
Zhao et al., 2013a). Annual average $SO_2$ concentrations in Shanghai were in the
range of 45-61 µg m$^{-3}$ during 2000-2005 and decreased by around 50% to 24-29 µg
m$^{-3}$ during 2010-2013 (http://www.zhb.gov.cn/). Note that $SO_2$ emissions in
Shanghai only accounted for less than 20% of the total $SO_2$ emissions in YRD and
with small annual variations. The high concentrations of $SO_4^{2-}$ observed in Shanghai
were also closely related to regional transport from north China (e.g. BTH and
Shandong province) (Li et al., 2011; Wang et al., 2016a).
Annual $NO_3^-$ concentrations in Shanghai were relatively steady (6.0-7.7 µg m$^{-3}$)
during 1999-2009, but sharply increased to 13.3 µg m$^{-3}$ in 2011-2014. $NO_x$
emissions in YRD also showed an increasing trend during these years, consistent
with satellite retrieved vertical column $NO_2$ densities during 2000-2010 (Zhang et al.,
2012b; Zhao et al., 2013b). In contrast, surface-level annual $NO_2$ concentrations in
Shanghai sharply decreased from 90 µg m$^{-3}$ in 2000 to a range of 48-61 µg m$^{-3}$
during 2003-2013 (http://www.zhb.gov.cn/). The inconsistency in the trends between
emissions and gaseous and particulate surface air concentrations was also found in
Beijing. Photochemistry and regional transport of related pollutants should be the
major causes of this phenomenon.
Annual $NH_4^+$ concentrations decreased from 5.9 µg m$^{-3}$ in 1999-2000 to the
levels of 4.1 µg m$^{-3}$ in 2009 and then increased to 6.6 µg m$^{-3}$ in 2011-2014. $NH_3$
emissions increased in 2000-2005 in east China (including BTH, YRD and PRD) and
possibly also increased in 2006-2010 due to the lack of control measures for $NH_3$ in
China (Wang et al., 2011). The recently increased $NH_4^+$ concentrations in Shanghai
were likely due to the concurrent increases of $NH_3$ emissions and $NO_3^-$
concentrations.
In summary, a decreasing trend was identified in TC, increasing ones for $NO_3^-$
and $NH_4^+$, and a stable one for $SO_4^{2-}$ in Shanghai. The inter-annual variations in EC
agreed with its local emission trends in Shanghai rather than regional emission trends.
In contrast, inter-annual variations in $SO_4^{2-}$, $NO_3^-$ and $NH_4^+$ agreed more with the
regional scale emission trends of their respective gaseous precursors in YRD. Similar
to Beijing, nonlinear responses of concentration changes of these aerosol
components to their respective emission trends were also found in Shanghai.
**2.2.2.2 Relative contributions to PM$_{2.5}$**
The chemical compositions in PM$_{2.5}$ between the cities in YRD were compared
between Shanghai and Nanjing due to the lack of continuous annual data in
Hangzhou. A conversion factor of 1.6 between OC and OM was chosen for YRD,
slight smaller than that (1.8) chosen for BTH considering the lower impact of
biomass burning on PM$_{2.5}$ in this region (Feng et al., 2006; Li et al., 2016a).
Secondary inorganic aerosols contributed 25-54% of PM$_{2.5}$ annually in Shanghai and
Nanjing, while carbonaceous aerosols contributed 28-47% (Li et al., 2016a; Wang et
al., 2016a; Ye et al., 2003; Zhao et al., 2015b). The sum of secondary inorganic
aerosols (sulfate, nitrate and ammonium) and carbonaceous aerosols (OM and EC)
accounted for 76% and 66% of $PM_{2.5}$ mass in Shanghai and Nanjing, respectively,
which was comparable with those (65%-70%) in BTH.

Seasonal variations of secondary inorganic aerosols contributions to $PM_{2.5}$ were

small in both cities, e.g., 41-49% in Shanghai and 32-40% in Nanjing (Fig.5). Larger
seasonal variations were found for carbonaceous aerosols than secondary inorganic
aerosols, e.g., 47% in summer and 33%-39% in other seasons in Shanghai, and
ranged from 27% (spring) to 65% (autumn) in Nanjing (Cao et al., 2012b; Huang et
al., 2014a; Huang et al., 2014b; Li et al., 2016a; Ming et al., 2017; Pathak et al., 2011;
Shen et al., 2014; Wang et al., 2016a; Yang et al., 2005b; Ye et al., 2003; Zhao et al.,
2015a).

In Hangzhou, seasonal contributions can only be estimated for summer and

winter 2003 (Cao et al., 2012b). Seasonal contribution of secondary inorganic
aerosols in winter was 44%, which was evidently higher than that in summer (34%),
while carbonaceous aerosols contributed 33-35%. At the rural sites (Ningbo and
Lin'an) in Zhejiang province, seasonal contributions of carbonaceous aerosols varied
within a small range (28%-34%) in four seasons in 2008-2009, which were
comparable with those in Hangzhou (Feng et al., 2015; Liu et al., 2013a).

In summary, the different seasonal average contributions of secondary inorganic

aerosols and carbonaceous aerosols in Shanghai and Nanjing were likely due to
different local sources in YRD. The seasonal patterns of these chemical components
in Shanghai were a result of both local emissions and regional transport, but in
Nanjing, the seasonal pattern was mainly determined by local emissions because it is
an inland city surrounded by many industrial enterprises including power plants,
petrochemical plants, and steel plants.
**2.2.3 The Pearl River Delta region**
**2.2.3.1 Inter-annual variations in Guangzhou**
Inter-annual variations for dominant chemical components were only discussed
for Guangzhou in PRD since data for this city were available during 2002-2003,
2009-2010 and 2014 (Hagler et al., 2006; Tao et al., 2014c; Tao et al., 2017). Data
for Shenzhen were only available during 2002-2003 and 2009 (Hagler et al., 2006;
Huang et al., 2013) and for Hong Kong during 2000-2001 and 2002-2003 (Hagler et
al., 2006; Louie et al., 2005b). Annual OC concentrations decreased significantly
from 17.6 µg m$^{-3}$ in 2002-2003 to 9.0 µg m$^{-3}$ in 2009-2010, and then to 8.2 µg m$^{-3}$ in
2014 in Guangzhou, while EC slightly increased from 4.4 µg m$^{-3}$ to 6.0 µg m$^{-3}$ and
then decreased to 4.0 µg m$^{-3}$ during the same periods. Similar to Guangzhou, annual
OC concentrations decreased significantly from 11.1 µg m$^{-3}$ in 2002-2003 to 8.3 µg
m$^{-3}$ in 2009-2010 in Shenzhen, while EC slightly increased from 2.3 µg m$^{-3}$ to 2.7 µg
m$^{-3}$. Apparently, the trends of EC in Guangzhou and Shenzhen were inconsistent
with the BC emission trends in Guangdong province during 2005-2009, which
showed a slightly decreasing trend (Qin and Xie, 2012). As a result, TC
concentrations gradually decreased from 22.0 µg m$^{-3}$ to 15.0 µg m$^{-3}$ in Guangzhou
and from 15.0 µg m$^{-3}$ to 13.0 µg m$^{-3}$ in Shenzhen before 2010, similar to Beijing and
Shanghai. The reduction of TC was significant in Guangzhou (32%). The same
phenomenon was also observed at a suburban site of Guangzhou (Hagler et al., 2006;
Lai et al., 2016).

In contrast to the TC trend, annual $SO_4^{2-}$, $NO_3^-$ and $NH_4^+$ concentrations in

Guangzhou increased from 14.7, 4.0 and 4.5 µg m$^{-3}$ in 2002-2003 to 18.1, 7.8 and
5.1 µg m$^{-3}$ in 2009-2010 and decreased to 9.3, 2.2 and 3.8 µg m$^{-3}$ in 2014,
respectively. Increases in the concentrations were also found in Shenzhen, e.g., from
10.0, 2.3 and 3.2 µg m$^{-3}$ in 2002-2003 to 11.7, 2.7 and 3.5 µg m$^{-3}$ in 2009,
respectively (Hagler et al., 2006; Huang et al., 2013), and in the suburban areas of
Guangzhou, e.g., from 10.4, 0.3 and 2.4 µg m$^{-3}$ in 2002-2003 to 12.2, 2.0 and 5.2 µg
m$^{-3}$ in 2012-2013, respectively (Hagler et al., 2006; Lai et al., 2016). $SO_2$ emissions
in Guangdong province gradually increased in the previous decade, e.g., 964, 1150,
1177 and 1258 Gg year$^{-1}$ in 2000, 2004, 2007 and 2010, respectively (Lu et al., 2010;
Zhao et al., 2013a). However, $SO_2$ emissions in PRD decreased by more than 40%
between 2005 and 2009, due to flue gas desulfurization facilities in power plants and
large industrial boilers installed in this region (Lu et al., 2013). Annual average $SO_2$
concentrations in Guangzhou gradually increased from 45 µg m$^{-3}$ in 2000 to 77 µg
m$^{-3}$ in 2004, and then decreased to 17 µg m$^{-3}$ in 2014 (http://www.gzepb.gov.cn/).
Thus, the increased $SO_4^{2-}$ concentration before 2010 in Guangzhou was largely due
to regional transport of pollutants from outside of PRD. The decreased $SO_4^{2-}$
concentration in 2014 in Guangzhou was likely due to flue gas desulfurization
facilities in power plants and large industrial boilers implemented across the entire
Guangdong province (http://www.gdep.gov.cn/).
Meanwhile, $NO_x$ emissions increased in Guangdong province as well as across
the entire PRD, similar to the trends in BTH and YRD (Lu et al., 2013; Zhang et al.,
2012b; Zhao et al., 2013b). However, annual average surface $NO_2$ concentrations in
Guangzhou fluctuated from 61 to 73 µg m$^{-3}$ during 2000-2007 and from 48 to 56 µg
m$^{-3}$ during 2008-2014 (http://www.gzepb.gov.cn/). An opposite trend was also found
between $NO_2$ and $NO_x$ emissions with the former persistently decreased while the
latter increased in Guangzhou, although $NO_3^-$ concentrations also increased. Thus,
emissions as well as chemical processes both affected these ions concentrations in air.
Annual $NH_4^+$ concentrations slightly increased about 10% before 2010 in
Guangzhou and Shenzhen although $NH_3$ emissions changed little during 2002-2006
in PRD (Zheng et al., 2012). Thus, the slightly increased $NH_4^+$ concentrations, if not
caused by measurement uncertainties, in Guangzhou and Shenzhen during
2002-2010 were largely due to the increased $SO_4^{2-}$ and $NO_3^-$, which enhanced the
conversion of $NH_3$ to $NH_4^+$.
In summary, a decreasing trend was identified in TC and increasing trends were
found for $SO_4^{2-}$, $NO_3^-$ and $NH_4^+$ in Guangzhou and Shenzhen before 2010, while all
chemical components decreased after 2010 in Guangzhou. The inter-annual
variations in EC were inconsistent with BC emission trends in Guangdong province.
In contrast, inter-annual variations in $SO_4^{2-}$, $NO_3^-$ and $NH_4^+$ agreed with regional
scale emission trends of their respective gaseous precursors in Guangdong province
rather than PRD. Similar to Beijing and Shanghai, nonlinear responses of
concentration changes of these aerosol components to their respective emission
trends were also found in Guangzhou and Shenzhen.

**2.2.3.2 Relative contributions to PM$_{2.5}$**

Data collected in 2002-2003 were discussed since multiple cities (e.g.
Guangzhou, Conghua, Zhongshan, Shenzhen and Hong Kong) in PRD have data
during this period  (Hagler et al., 2006). The conversion factor between OC and
OM was chosen to be the same as in YRD (1.6). Secondary inorganic aerosols
contributed 33-38%, depending on location, of PM$_{2.5}$ annually, while carbonaceous
aerosols contributed 37-46%. It is noted that PM$_{2.5}$ in Guangzhou was much higher
than those in other coastal cities (including Zhongshan, Shenzhen and Hong Kong),
but the contributions of secondary inorganic aerosols and carbonaceous aerosols
were not significantly different between these cities. At rural sites (Tianhu and
Conghua near Guangzhou and Hok Tsui near Hong Kong), secondary inorganic
aerosols and carbonaceous aerosols accounted for 35-48% and 24-43%, respectively,
of the PM$_{2.5}$ mass, which were similar to those obtained in the cities in PRD (Hagler
et al., 2006; Lai et al., 2016; Louie et al., 2005b). Thus, the sum of secondary
inorganic aerosols and carbonaceous aerosols accounted for 68%-83% of the PM$_{2.5}$
mass in the PRD region, similar to Shanghai (YRD).
Although many studies have been conducted in PRD, many of them were short
term. Studies covering all four seasons were mainly carried out in Guangzhou and
Hong Kong (Fig.6) (Andreae et al., 2008; Cao et al., 2004; Cao et al., 2012b; Cui et
al., 2015; Ho et al., 2006a; Huang et al., 2014b; Jung et al., 2009; Lai et al., 2007;
Liu et al., 2014a; Louie et al., 2005a; Tan et al., 2009; Tao et al., 2009; Tao et al.,
2014c; Tao et al., 2015b; Tao et al., 2017; Yang et al., 2011b). Seasonal average
contributions of secondary inorganic aerosols were generally higher in spring and
autumn than in summer and winter in both Guangzhou and Hong Kong. If all the
years of data were averaged together, secondary inorganic aerosols contributed 43%,
31%, 38% and 33% in spring, summer, autumn and winter, respectively, to $PM_{2.5}$ in
Guangzhou and 45%, 25%, 46% and 37%, respectively, in Hong Kong. However,
different seasonal patterns were found between Guangzhou and Hong Kong for
carbonaceous aerosols. Carbonaceous aerosols contributed 34%, 37%, 35% and 34%
in spring, summer, autumn and winter, respectively, to $PM_{2.5}$ in Guangzhou and 54%,
47%, 49% and 38%, respectively, in Hong Kong. Seasonal variations of OC/EC
ratios ranged from 1.6 to 3.4 in Guangzhou and from 1.2 to 2.1 in Hong Kong,
suggesting coal combustion and vehicle exhaust were the dominant sources in
Guangzhou while vehicle exhaust was the dominant source in Hong Kong (He et al.,
2008; Watson et al., 2001).
**2.2.4 Other cities**
Besides the cities in BTH, YRD and PRD, synchronous measurements of $PM_{2.5}$
and the dominant chemical components have also been conducted in several cities in
other regions of China, mainly in the capital city of a province (e.g. Zhengzhou of
Henan province (Geng et al., 2013), Xi'an of Shaanxi province (Wang et al., 2015c),
Lanzhou of Gansu province (Wang et al., 2016b), Jinan of Shandong province (Yang
et al., 2012), Chengdu of Sichuan province (Tao et al., 2013a; Tao et al., 2014b),
Chongqing of Chongqing municipality (Yang et al., 2011b), Changsha of Hunan
province (Tang et al., 2017), Xiamen and Fuzhou of Fujian province (Xu et al.,
2012b; Zhang et al., 2016) and Haikou of Hainan province (Liu et al., 2017a)). A
conversion factor of 1.6 between OC and OM was chosen for Fuzhou, Xiamen and
Haikou and 1.8 for other cities based on their geographical locations.
Annual average contributions of secondary inorganic aerosols and carbonaceous
aerosols to $PM_{2.5}$ were 30% and 36%, respectively, in the island city Haikou, similar
to what was found in Beijing, and were 43-46% and 29-36%, respectively, in the
coastal cities Fuzhou and Xiamen, similar to what was found in Shanghai. Annual
contributions of secondary inorganic aerosols ranged from 29% to 39% in inland
cities (Zhengzhou, Xi'an, Jinan, Chengdu, Chongqing and Changsha) except
Lanzhou (15%), which were comparable with those observed in PRD (33-41%). In
contrast, large differences were found in the annual contributions of carbonaceous
aerosols, ranging from 23% in Zhengzhou to 47% in Chongqing. The sum of
secondary inorganic aerosols and carbonaceous aerosols accounted for 56%-79% of
the $PM_{2.5}$ mass in these cities.
At an Asian continental outflow site (Penglai in Shandong province), annual
average contribution of secondary inorganic aerosols to $PM_{2.5}$ reached 54% (Feng et
al., 2012b), evidently higher than those in urban and inland rural sites in China,
while that of carbonaceous aerosols was 31%, close to those in BTH. This finding
suggested that intensive emissions of $SO_2$ and $NO_x$ in China enhanced the
downward transport of secondary inorganic aerosols to the Pacific Ocean.
Seasonal average contributions are only shown here for Jinan (Yang et al., 2012),
Zhengzhou (Geng et al., 2013), Fuzhou (Xu et al., 2012b), Chengdu (Tao et al.,
2013a; Tao et al., 2014b), Lanzhou (Tan et al., 2016b; Wang et al., 2016b), Xiamen
(Zhang et al., 2012a), Changsha (Tang et al., 2017), and Haikou (Liu et al., 2017a)
due to the incomplete data in Xi'an and Chongqing (Fig. 7). Seasonal contributions
of secondary inorganic aerosols were evidently higher in summer than in other
seasons in Zhengzhou, Jinan and Lanzhou (typical northern cities), similar to what
was seen in BTH. In the southwest city Chengdu and the central city Changsha,
seasonal contribution of secondary inorganic aerosols in spring was only 30% and
27%, respectively, lower than other seasons (40-42% and 30-31%, respectively). In
the two southern coastal cities Fuzhou and Xiamen, the highest seasonal average
contribution of secondary inorganic aerosols was observed in winter (53% and 33%,
respectively), much higher than in other seasons (34-42% and 21-24%, respectively).
In the southern island city Haikou, seasonal contributions of secondary inorganic
aerosols were also slightly higher in winter (30%) than in other seasons (21-27%),
similar to the coastal cities Fuzhou and Xiamen.

Seasonal average contributions of carbonaceous aerosols were evidently higher

in cold seasons than in warm seasons in the three northern cities (Zhengzhou and
Jinan and Lanzhou) due to heating activities and biomass burning, similar to BTH.
Surprisingly, a similar seasonal pattern was also found in one coastal city Xiamen,
e.g., 38% in winter versus 27-30% in other seasons. In contrast, higher seasonal
contributions were found in warm season than in cold seasons in the southern coastal
city (Fuzhou) and the southern island city (Haikou). No seasonal variations were
found in the southwest inland city Chengdu (29%-32%) and the central inland city
Changsha (28-33%). The summed contributions of secondary inorganic aerosols and
carbonaceous aerosols were evidently lower in spring than in other seasons in most
of the northern cities (e.g. Jinan, Lanzhou, Zhengzhou, and BTH), likely due to the
frequent spring dust storm events in northern China.
**2.2.5 Summary of PM$_{2.5}$ chemical properties**
Carbonaceous aerosols showed decreasing trends over the last ten years
(2000-2010) in Beijing, Shanghai and Guangzhou, consistent with BC emission
trends in these cities and surrounding areas. $SO_4^{2-}$ and $NH_4^+$ remained at high levels
with no significant trends in Beijing and Shanghai, but with an increasing trend in
Guangzhou. $NO_3^-$ showed increasing trends in all of the above-mentioned megacities.
Annual mass concentrations of PM$_{2.5}$, secondary inorganic aerosols, and
carbonaceous aerosols showed similar spatial gradients decreasing from high to low
latitude regions.
Annual average contributions of secondary inorganic aerosols to PM$_{2.5}$ ranged
from 25% to 48% with higher values in southern regions, and those of carbonaceous
aerosols ranged from 23% to 47%, also with higher values in southern regions
(Fig.8). The percentage contributions of the sum of secondary inorganic aerosols and
carbonaceous aerosols were higher in southern cities than in northern cities due to
the frequent dust events in the north.
The highest seasonal average contributions of secondary inorganic aerosols to
PM$_{2.5}$ were observed in summer in most of the northern cities, but can occur in
different seasons in southern cities. In contrast, the highest seasonal contributions of
carbonaceous aerosols were observed in cold seasons in most of the northern cities,
and in warm seasons in most of the southern cities. The different seasonal patterns
were largely caused by heating and biomass burning in cold seasons in north China.
**2.3 Source apportionment of PM$_{2.5}$**
Advantages of receptor-based methods used for source apportionment analysis
for various pollutants were discussed in Cheng et al. (2015) and Hopke (2016).
Source apportionment studies of PM$_{2.5}$ in China using receptor models have also
been reviewed recently covering a wide range of topics (Liang et al., 2016; Lv et al.,
2016; Pui et al., 2014; Zhang et al., 2017a). However, a general summary of
spatial-temporal patterns of PM$_{2.5}$ source factors and their relative contributions is
still lacking, which is the focus of the discussion below. Data collected in this section
are listed in Table S2 of the SI document.
Commonly used receptor models in source apportionment of PM$_{2.5}$ in China
include Principal Component Analysis/Absolute Principal Component Scores
(PCA/APCS), Chemical Mass Balance receptor (CMB), Positive Matrix
Factorization (PMF), and UNMIX and Multilinear Engine-2 model (ME-2). Among
these, PMF and CMB models were the most widely used in China. Quantitative
assessments of the uncertainties in using these methods are rare; studies using the
same dataset collected in 2000 in Beijing and applying the above-mentioned models
suggested that, while the models still identified the same dominant source factors,
the relative contributions from these source factors differed by as much as 30%
between the different models (Song et al., 2006a, b). Similar magnitudes of
uncertainties could also be caused by using different biomass burning tracers despite
using the same receptor model (Tao et al., 2016b).
Major source factors identified for $PM_{2.5}$ in most Chinese cities include
secondary inorganic aerosols (SIA), coal combustion (COAL), biomass burning
(BIOM), traffic emission (TRAF), dust emission (DUST), and industrial emission
(INDU). Other source factors were also identified (and sometimes due to using more
specific source names), such as metal manufacturing (including iron and steel
industry, Cu smelting) in industrial cities (e.g. Dongying and Tai'an of Shandong
province, Nanjing, Hangzhou, Lanzhou, Chengdu, Chongqing and Changsha), and
sea salt and ship emissions in coastal cities (e.g. Longkou of Shandong province,
Nanjing, Guangzhou, Zhuhai and Hong Kong). Contributions of dominant source
factors to $PM_{2.5}$ are discussed below in detail on a regional basis.
**2.3.1 The Beijing-Tianjin-Hebei region**
Studies in Beijing covered multiple years and mostly used the PMF model. If
averaging the results from the years in 2000 (Song et al., 2006b), 2001-2004 (Zhang
et al., 2007), 2009-2010 (Zhang et al., 2013a), and 2012-2013 (Zíková et al., 2016),
the six source factors (SIA, COAL, BIOM, TRAF, DUST, and INDU) accounted for
31±12%, 16±4%, 12±1%, 16±13%, 12±7%, and 9±11%, respectively, of the $PM_{2.5}$
mass in Beijing. There was an increasing trend for SIA contributions (from 19% to
48%), a decreasing trend for COAL (from 19% to 11%), and a stable trend for BIOM
(11-12%) during 2000-2013. There was more uncertainties in identifying TRAF and
INDU than other source factors due to the differences in the source profiles.
A study in Tianjin in 2013-2014 only identified four dominant sources (SIA,
COAL, TRAF, and DUST) using Multilinear Engine-2 model (ME-2), which
accounted for 41%, 25%, 14%, and 20%, respectively, of the annual average $PM_{2.5}$
mass (Tian et al., 2016). Compared with results in 2012-2013 in Beijing (Ziková et
al., 2016), the contributions of SIA were comparable in the two cities, but those of
COAL and DUST were much higher in Tianjin than Beijing. However, the results
from an earlier study in Tianjin in 2009-2010 were much more comparable to those
in Beijing during the same years in terms of $PM_{2.5}$ levels and source attributions
(Zhao et al., 2013c), implying faster decrease of COAL contribution in Beijing than
Tianjin.
Seasonal results of source apportionment analysis are also available for Beijing
(Huang et al., 2014b; Song et al., 2007; Wu et al.,2014; Zheng et al., 2005; Zhang et
al., 2013a; Ziková et al., 2016). In most cases, SIA was the largest contributor in
spring, summer and autumn, accounting for 26-61% of the $PM_{2.5}$ mass, while COAL
was the largest contributor in winter, accounting for 13-57% of the $PM_{2.5}$ mass. The
contributions of the other sources were lower than those of SIA and COAL, but
subject to seasonal variations. For example, the largest seasonal contribution of
BIMO was in autumn and of DUST in spring.
**2.3.2 The Yangtze River Delta region**
Studies for one year or longer were only made in Nanjing (Li et al., 2016a) and
Hangzhou (Liu et al., 2015). Metal manufacturing was identified as a source factor in
both Nanjing and Hangzhou, while ship emissions were also identified in Nanjing.
Annual contributions of SIA to $PM_{2.5}$ mass reached 68% in Nanjing while all the
other sources (COAL, DUST, sea salt and ship emissions, and metal manufacturing)
each contributed 10% or less. In contrast, metal manufacturing, SIA, TRAF, and
COAL accounted for 32%, 28%, 17%, and 13%, respectively, of the $PM_{2.5}$ mass in
Hangzhou. Evidently, the contributions of SIA in Nanjing were much higher than
those in Hangzhou and cities in BTH. The contributions of COAL in Nanjing and
Hangzhou were similar, but were evidently lower than those in cities in BTH.

Similar to the cities in BTH, the largest seasonal average contribution of SIA in

Nanjing was in summer and of COAL in winter. Only winter data was available in
Shanghai (Huang et al., 2014b), and the contributions of SIA and DUST to $PM_{2.5}$
were similar between Shanghai and Nanjing.
**2.3.3 The Pearl River Delta region**

Studies covering one year or longer were available in Guangzhou (Tao et al.,

2017), Shenzhen (Huang et al., 2013), Hong Kong (Guo et al., 2009a), and suburban
Zhuhai (Tao et al., 2017) and suburban Hong Kong (Huang et al., 2014c). On an
annual basis, SIA contributed 50% to $PM_{2.5}$ mass in Guangzhou while other sources
(ship emissions, COAL, TRAF, and DUST) each contributed 7-17%. In Shenzhen,
SIA, TRAF and BIOM accounted for 39%, 27%, and 10%, respectively, of the $PM_{2.5}$
mass. In Hong Kong, SIA, TRAF, oil residue (related to Ni and V, or ship emissions),
DUST, and sea salt accounted for 28%, 23%, 19%, 10%, and 7%, respectively, of the
$PM_{2.5}$ mass.

735   Slightly different sources factors were identified in suburban studies. Annual

736 contributions from mixed source (from regional transport), secondary nitrate and

737 chloride, ship emissions, COAL, and electronic industries accounted for 36%, 20%,

738 18%, 13%, and 13%, respectively, of the $PM_{2.5}$ mass in suburban Zhuhai, while SIA,

739 BIOM, sea salt, residual oil combustion (related to Ni and V, or ship emission),

740 DUST, and TRAF accounted for 39%, 20%, 17%, 12%, 7%, and 5%, respectively, of

741 the $PM_{2.5}$ mass in suburban Hong Kong.

742   Despite the slightly different source factors identified between urban and

743 suburban sites in PRD, SIA was the largest contributor to $PM_{2.5}$ mass in this region.

744 Ship emissions were identified in this region, but not in northern China, and this

745 source factor contributed more than 10% of the $PM_{2.5}$ mass in all the studies except

746 the one for Shenzhen. Similar to the cities in northern China, the high contribution

747 from coal combustion was also found in Guangzhou and suburban Zhuhai.

748   Seasonal results of source apportionment analysis were available for four seasons

749 in suburban Hong Kong (Huang et al., 2014c), winter in Guangzhou (Huang et al.,

750 2014b), and summer and winter in Foshan (Tan et al., 2016c) and Hong Kong (Ho et

751 al., 2006b). SIA was the largest contributor to $PM_{2.5}$ among all the identified source

752 factors in every season in suburban Hong Kong (30-45%) and in winter in

753 Guangzhou (59%). In contrast, INDU was the largest contributor in winter in Foshan

754 (39%), a typical industrial city in PRD (Tan et al., 2016c). In suburban Hong Kong,

755 seasonal average contribution of SIA was the lowest in summer, different from what

756 was found for cities in BTH and YRD, while that of sea salt and ship emissions were

the highest in summer due to the prevailing air masses from the South China Sea
(Huang et al., 2014c).
**2.3.4 Other cities**
Studies covering one year or longer were mostly conducted for provincial capital
cities including Jinan (Yang et al, 2013), Zhengzhou (Geng et al., 2013), Xi'an
(Wang et al., 2015c), Lanzhou (Wang et al, 2016b), Chengdu (Tao et al., 2014a),
Chongqing (Chen et al., 2017), Changsha (Tang et al., 2017), Wuhan (Xiong et al.,
2017), Xiamen (Zhang et al., 2016) and Haikou (Liu et al., 2017a), and for an inland
city Heze (Liu et al., 2017b) and a regional background site (located in Yellow River
Delta National Nature Reserve in Dongying city) (Yao et al., 2016) both in Shandong
province. Annual results were available from most studies, but were aggregated from
seasonal results for Wuhan and Haikou. All the sites were grouped into four regions
for easy discussion, i.e., northwest China (Lanzhou and Xi'an), southwest China or
Sichuan basin (Chengdu and Chongqing), eastern and central China (Jinan,
Zhengzhou, Heze, Dongying, Wuhan and Changsha), and south coastal cities
(Xiamen and Haikou).
The two northwest cities showed the same top four dominant source factors,
although with slightly different percentage contributions to $PM_{2.5}$ mass, e.g., 29%
from SIA, 19% from COAL, 17% from DUST, and 15% from TRAF in Xi'an, and
17% from SIA and 22% from the other three sources in Lanzhou. The lower SIA
contribution in Lanzhou was likely due to the dry climate inhibiting formation of
SIA. Similar results to those in Xi'an were also obtained in rural Xi'an, with SIA,
COAL, DUST and TRAF contributing 31%, 16%, 20% and 13%, respectively, to the
PM$_{2.5}$ mass (Wang et al., 2015c). The two southwest cities (Chengdu and Chongqing)
showed nearly the same source-apportionment analysis results with SIA contributing
just below 40% and COAL and INDU each contributing around 20% to PM$_{2.5}$ mass.
The same top four dominant source factors (SIA, COAL, DUST and TRAF)
found in the two northwest cities were also found in other capital cities (Jinan,
Zhengzhou, and Wuhan, except Changsha) and a medium size city (Heze) in eastern
and central China, which accounted for 24-55%, 14-23%, 5-26% and 5-27% of the
PM$_{2.5}$ mass, depending on location. In Changsha, SIA, mixed source of INDU and
BIOM, and DUST accounted for 60%, 27%, and 13%, respectively, to PM$_{2.5}$ mass.
SIA, BIOM and INDU were the most important sources, accounting for 54%, 16%,
and 16%, respectively, of PM$_{2.5}$ mass in a regional background site in Dongying.
Similar source-apportionment results were found between the two south coastal
cities (Xiamen and Haikou) with SIA, TRAF, DUST, COAL, and sea salt accounting
for 20-27%, 16-21%, 12-22%, 8-9% and 6-10%, respectively, of the PM$_{2.5}$ mass.
Seasonal results of source apportionment analysis are available for four seasons
in Jinan (Yang et al., 2013), a regional site in Dongying (Yao et al., 2016), Chengdu
(Tao et al., 2014a), Chongqing (Chen et al., 2017), Wuhan (Xiong et al., 2017) and
Haikou (Liu et al., 2017a), for summer and autumn in Tai'an of Shandong province
(Liu et al., 2016a) and Xi'an (Xu et al., 2016), for summer and winter in Lanzhou
(Tan et al., 2017), and for winter in Longkou (a coastal site in Shandong province)
(Zong et al., 2016).
In most seasons, SIA was the largest contributor to $PM_{2.5}$ mass, e.g., in Jinan
(30-45%), Tai'an (27%), a regional site in Dongying (35-72%), Chengdu (33-44%),
Chongqing (24-52%), Wuhan (23-41%), Lanzhou (15-33%), and Haikou (11-26%),
except during spring in Wuhan and summer in Haikou when DUST was the largest
contributor, during winter in Longkou and Xi'an when COAL was the largest
contributor, and during summer in Lanzhou when smelting industry was the largest
contributor. Only winter data was available in Longkou, and ship emissions
contributed 9% to $PM_{2.5}$ mass, similar to what was found in the cities of PRD.
**2.3.5 Summary of $PM_{2.5}$ source apportionment studies**
SIA, COAL and TRAF were the dominant source factors in most cities in China.
On an annual average, the sum of these three factors accounted for 63%-80% of
$PM_{2.5}$ mass in the cities of BTH region, 58%-78% in the cities of YRD region,
51%-67% in the cities of PRD region, 51%-61% in the northwest cities, 57%-60% in
the southwest cities, 57%-82% in the eastern and central cities, and 44%-57% in the
south coastal cities. The contributions of DUST were significant (7-26%) in northern
cities and a central city (Zhengzhou), of INDU significant (19-27%) in typical
industrial cities (e.g. Chengdu, Chongqing, Changsha), and of ship emission
significant (7-19%) in coastal and river cities (e.g. Longkou, Nanjing, Guangzhou,
Zhuhai, Hong Kong). High seasonal contributions were found for SIA in summer,
COAL in winter, DUST in spring, and ship emission in summer in applicable cities.
It should be noted that SIA chemical compounds are formed by reactions in the
atmosphere involving primary emissions of gaseous precursors that can be produced
from any of the identified sources factors discussed above as well as from sources
seldom mentioned in source apportionment studies such as agricultural emissions
and many natural sources. However, sources of the gaseous precursors are often
undetermined in source apportionment studies resulting in a large proportion of
$PM_{2.5}$ that cannot be explained (Karagulian et al., 2015). If the SIA contributions
can be allocated to specific types of primary emissions, the overall percentage
contributions from each of the identified source factor should be much higher,
especially for COAL, TRAF, INDU and BIMO due to their high emission rates of
primary pollutants of gaseous species. To identify the various types of primary
emission sources, datasets containing trace element and other chemical markers need
to be included in source apportionment models (Lee and Hopke, 2006). Combining
receptor-based analysis results with source-based studies using chemical transport
models can provide a more complete picture by quantifying contributions of
dominant emission sources to $PM_{2.5}$ pollution.
**3. Aerosol optical properties**
There were much fewer measurements of aerosol optical properties than
chemically resolved $PM_{2.5}$ data in China. Data reviewed in this section are all listed
in Table S3 of the supplement document. Measurements were available at urban sites
including Beijing in BTH (Bergin et al., 2001; Garland et al., 2009; Han et al., 2014;
He et al., 2009; Jing et al., 2015; Liu et al., 2009; Tian et al., 2015; Tao et al., 2015a;
Wu et al., 2016; Zhao et al., 2011), Shanghai (Cheng et al., 2015; Feng et al., 2014;
Han et al., 2015; Huang et al., 2014a; Li et al., 2013a; Xu et al., 2012a; Zha et al.,
2014), Nanjing (Kang et al., 2013), and Shouxian (Anhui province) in YRD (Fan et
al., 2010), Guangzhou, Shenzhen and Hong Kong in PRD (Andreae et al., 2008;
Cheng et al., 2006a; Cheng et al., 2006b; Cheng et al., 2008a; Gao et al., 2015;
Garland et al., 2008; Jung et al., 2009; Lan et al., 2013; Man and Shih, 2001; Tao et
al., 2014c; Verma et al., 2010; Wu et al., 2009; Wu et al., 2013), Chengdu in
southwest China (Tao et al., 2014b; Wang et al., 2017a), and Xi'an in northwest
China (Cao et al., 2012a; Zhu et al., 2015), rural sites including rural Beijing
(Shangdianzi) and rural Tianjin (Wuqing) in BTH (Ma et al., 2011; Yan et al., 2008;
Zhao et al., 2011), and remote sites in north and northwest China (Li et al., 2010; Xu
et al., 2004; Yan, 2007). Sites with one year or longer data included Beijing, rural
Beijing, Shanghai, Guangzhou, Chengdu, Xi'an and Shouxian.
Aerosol optical depth (AOD), representing the integrated light extinction
coefficient in a vertical column, can be obtained from MODerate-resolution Imaging
Spectroradiometer (MODIS) data. Satellite retrievals of AOD have been widely
applied to estimate surface $PM_{2.5}$ concentrations using statistical models (Liu et al.,
2005; Hu et al., 2013; Ma et al., 2014; Wang and Christopher, 2003). Although the
correlation between AOD and $PM_{2.5}$ mass concentration depends on many factors,
such as aerosol size distribution, refractive index, single-scattering albedo, and
meteorological factors (Che et al., 2009; Guo et al., 2009b; Guo et al., 2017), the
predicted $PM_{2.5}$ mass from satellite AOD data compared well with ground-level
measurements (Ma et al., 2014; Xie et al., 2015b). Moreover, the spatial distributions
of AOD measured using sun photometers mostly agreed with those retrieved from
satellite data (Che et al., 2014; Che et al., 2015; Liu et al., 2016b; Pan et al., 2010).
Spatial distributions of annual average AOD in 2014 are shown in Fig. 11. The
spatial distributions of $PM_{2.5}$ shown in Fig. 2 are similar to the patterns of AOD
shown in Fig. 11. Differences in fine structures of their patterns were due to surface
$PM_{2.5}$ versus column AOD comparison and spatial variations in $PM_{2.5}$ chemical
composition.
In this section, geographical patterns of the aerosol optical properties including
$b_{sp}$ and $b_{ap}$ measured on ground base in major Chinese cities are first discussed
(section 3.1). Temporal patterns of $b_{sp}$ and $b_{ap}$ on annual and seasonal scales are then
discussed for major regions (section 3.2). Fewer studies were available for $b_{ap}$ than
$b_{sp}$, however, the measured BC concentrations (at 880 nm wavelength) can be
converted to $b_{ap}$ (at 532 nm wavelength) by a factor of 8.28 $m^2$ $g^{-1}$ (Wu et al., 2009).

## 3.1 Geographical patterns

Annual average $b_{sp}$ and $b_{ap}$ from ground measurements in major cities in China
are plotted in Fig. 12. Most $b_{sp}$ measurements were conducted using a nephelometer
under RH<60%. The highest annual $b_{sp}$ was in Xi'an (525 $Mm^{-1}$, RH<60%) (Cao et
al., 2012a), followed by Chengdu (456 $Mm^{-1}$, RH<40%; 421 $Mm^{-1}$, ambient RH)
(Tao et al., 2014b; Wang et al., 2017a), Guangzhou (326 $Mm^{-1}$, RH<70%) (Tao et al.,
2014c), Beijing (309 $Mm^{-1}$, RH<60%) (He et al., 2009; Jing et al., 2015; Zhao et al.,
2011), and Shanghai (217 $Mm^{-1}$, RH<60%) (Cheng et al., 2015b). Such spatial
patterns were mainly due to the spatial patterns of annual $PM_{2.5}$ mass, i.e. Xi'an (177
$\mu g$ $m^{-3}$) > Chengdu (111 $\mu g$ $m^{-3}$) > Beijing (108 $\mu g$ $m^{-3}$) > Shanghai (77 $\mu g$ $m^{-3}$) >
Guangzhou (65 µg m$^{-3}$), and partly due to humidity conditions, e.g., Beijing versus
Guangzhou. Noticeably, $b_{sp}$ in Shouxian County was higher than those in several
megacities (e.g. Beijing, Shanghai and Guangzhou), suggesting hazy weather also
frequently occurred even in small cities in China (Fan et al., 2010). $b_{sp}$ in rural
Beijing was 179 Mm$^{-1}$ (Yan et al., 2008; Zhao et al., 2011), which was much lower
than that in urban Beijing, but was close to the level in Shanghai.
Annual average $b_{ap}$ ranged from 37 to 96 Mm$^{-1}$ with higher values observed in
Chengdu and Xi'an (likely due to popular biomass burning besides large amount of
coal burning) (Cao et al., 2012a; Tao et al., 2014a; Tao et al., 2014b; Wang et al.,
2017a; Zhang et al., 2014b), and lower values in Shouxian and rural Beijing (Fan et
al., 2010; Yan et al., 2008; Zhao et al., 2011). $b_{ap}$ in Guangzhou was higher than that
in Beijing and Shanghai despite their similar PM$_{2.5}$ EC levels, likely due to the
different coating of EC in Guangzhou than in other cities. For example, the mass
absorption of EC in Guangzhou was 8.5 m$^2$ g$^{-1}$ (at 532 nm) in autumn 2004 (Andreae
et al., 2008), which was higher than that (4.2 m$^2$ g$^{-1}$ at 870 nm, equivalent to 7.2 m$^2$
g$^{-1}$ at 532 nm) in winter 2013 in Beijing (Wu et al., 2016).

## 905 **3.2 Temporal patterns**

### 906 **3.2.1 The Beijing-Tianjin-Hebei region**

$b_{sp}$ measurements in BTH longer than one year were only available in Beijing,
including the years of 2005, 2006, 2008-2009 and 2009-2010 (He et al., 2009; Jing et
al., 2015; Zhao et al., 2011). Annual $b_{sp}$ in Beijing increased by 36% from 264 Mm$^{-1}$
in 2005 to 360 Mm$^{-1}$ in 2009-2010, while PM$_{2.5}$ increased by 20% from 107 to 129
µg m$^{-3}$ during the same period. However, annual b$_{ap}$ in 2009-2010 was 64 Mm$^{-1}$,
which was slightly higher than 56 Mm$^{-1}$ in 2005-2006, although the annual EC in
2009-2010 was evidently lower than that in 2005-2006. Meanwhile, annual
secondary inorganic aerosols in 2009-2010 were evidently lower than that in
2005-2006. The coating by secondary inorganic aerosols likely enhanced the
absorption of EC (Bond et al., 2006; Cheng et al., 2009; Yu et al., 2010).
b$_{sp}$ measurements in rural Beijing included the years 2003-2005 (175 Mm$^{-1}$) and
2008-2009 (182 Mm$^{-1}$), while b$_{ap}$ only included the years of 2003-2005 (18 Mm$^{-1}$)
(Yan et al., 2008; Zhao et al., 2011). Considering all of the above-mentioned data
together, we can conclude that b$_{sp}$ and b$_{ap}$ showed slightly increasing tendencies in
urban and rural Beijing in recent years.
Seasonal variations of b$_{sp}$ and b$_{ap}$ at urban and rural sites in Beijing are plotted in
Fig. 13. The highest seasonal average b$_{sp}$ in Beijing was observed in winter and the
lowest in spring with seasonal variations up to a factor of 1.7 (Bergin et al., 2001;
Garland et al., 2009; Han et al., 2014; He et al., 2009; Jing et al., 2015; Li et al.,
2013b; Liu et al., 2009; Tao et al., 2015a; Tian et al., 2015; Zhao et al., 2011). A
different seasonal pattern was seen at the rural site located north of Beijing, which
showed 10-26% higher values in summer than in the other seasons (Yan et al., 2008;
Zhao et al., 2011). The highest seasonal b$_{sp}$ in winter in Beijing was consistent with
the highest seasonal PM$_{2.5}$ mass. However, in rural Beijing the highest PM$_{2.5}$ mass
was observed in spring due to the frequent dust storm events, and the second highest
seasonal average PM$_{2.5}$ mass in summer which corresponded to the highest seasonal
$b_{sp}$. This is because scattering efficiency of dust aerosols was lower than that of
anthropogenic aerosols (Zhao et al., 2011).
The highest seasonal $b_{ap}$ in Beijing appeared in autumn and the lowest in spring
with seasonal variations up to a factor of 2.0 (Bergin et al., 2001; Garland et al., 2009;
He et al., 2009; Jing et al., 2015; Li et al., 2013c; Liu et al., 2009; Tian et al., 2015;
Wu et al., 2016). Seasonal variations of $b_{ap}$ were different from those of $b_{sp}$ due to
their dependence on different chemical compounds, i.e. $b_{sp}$ mainly on PM mass
while $b_{ap}$ mainly on EC mass in PM and its coating. In rural Beijing $b_{ap}$ was lower
by 19%~57% in summer than in other seasons, and with similar seasonal variations
to $b_{sp}$, suggesting aerosols in rural Beijing mainly came from regional transport (Yan
et al., 2008).
At the rural site in Tianjin (Wuqing) located between Beijing and Tianjin, only
spring and summer 2009 and winter 2010 data were available, which gave a seasonal
average of 280 $Mm^{-1}$ in spring, 379 $Mm^{-1}$ in summer, and 485 $Mm^{-1}$ in winter for $b_{sp}$,
and 47 $Mm^{-1}$ in spring and 43 $Mm^{-1}$ in summer for $b_{ap}$ (Fig. 13) (Chen et al., 2014b;
Ma et al., 2011). These seasonal values in Wuqing were higher than those observed
at the rural sites near Beijing, likely because Wuqing is close to and downwind of
Tianjin and Hebei province where major pollutant sources are located.
**3.2.2 The Yangtze River Delta and Pearl River Delta region**
No multi-year $b_{sp}$ measurement data were available for exploring inter-annual
variations, although multi-year measurements of BC or $b_{ap}$ were made in Shanghai
(YRD) and Guangzhou (PRD). Annual $b_{ap}$ in 2011-2012 (19 $Mm^{-1}$) was evidently
lower than that in 2010 (31 $Mm^{-1}$) in Shanghai (Feng et al., 2014; Zha et al., 2014),
consistent with the trend of EC, e.g. annual concentration of EC in 2012 (2.0 µg $m^{-3}$)
was only half of that in 2009 (4.1 µg $m^{-3}$) (Wang et al., 2016a; Zhao et al., 2015b). In
Guangzhou, annual $b_{ap}$ in 2007 (51 $Mm^{-1}$) was also evidently lower than that in 2004
(90 $Mm^{-1}$) (Wu et al., 2009), while EC in 2006-2007 (4.0 µg $m^{-3}$) was similar or
slightly lower than that in 2002-2003 (4.4 µg $m^{-3}$) (Hagler et al., 2006; Huang et al.,
2012). Thus, the inter-annual variations in $b_{ap}$ were mainly determined by EC trends
in the same cities.
$b_{sp}$ and $b_{ap}$ in winter were evidently higher than those in spring in Shanghai,
consistent with the seasonal patterns of $PM_{2.5}$ and EC, respectively (Fig. 13) (Cao et
al., 2012b; Cheng et al., 2015; Feng et al., 2014; Han et al., 2015; Huang et al.,
2014a; Li et al., 2013a; Pathak et al., 2011; Wang et al., 2016a; Xu et al., 2012a; Ye
et al., 2003; Zha et al., 2014; Zhao et al., 2015a). Similar seasonal variations were
found for $b_{sp}$ and $b_{ap}$ in the two PRD cities (Guangzhou and Hong Kong), which also
agreed with the patterns of $PM_{2.5}$ and EC (Andreae et al., 2008; Cao et al., 2004; Cao
et al., 2012b; Cui et al., 2015; Gao et al., 2015; Huang et al., 2014b; Jung et al., 2009;
Lai et al., 2007; Liu et al., 2014a; Louie et al., 2005a; Pathak et al., 2011; Tao et al.,
2009; Tao et al., 2014c; Tao et al., 2015b; Tao et al., 2017; Verma et al., 2010; Wu et
al., 2009; Wu et al., 2013). The highest $b_{sp}$ and $b_{ap}$ appeared in winter and the lowest
in summer with seasonal variations up to a factor of 3.1 and 17.1 for $b_{sp}$, 2.3 and 5.9
for $b_{ap}$, in Guangzhou and Hong Kong, respectively.

### 3.2.3 Other cities

In Chengdu of southwest China, the highest $b_{sp}$ appeared in winter and the lowest in summer with seasonal variations up to a factor of 1.9, which was consistent with the seasonal pattern of $PM_{2.5}$ (Tao et al., 2014a, b). However, the highest $b_{ap}$ appeared in spring despite the highest EC in winter (Tao et al., 2014b). One explanation could be due to the large amount of OC emitted from biomass burning in spring, which enhanced the absorption of EC (Schnaiter et al., 2005; Tao et al., 2013b). $b_{sp}$ and $b_{ap}$ in winter were evidently higher than those in summer in Xi'an in northwest China, consistent with the seasonal patterns of $PM_{2.5}$ and EC, respectively (Cao et al., 2009; Cao et al., 2012a; Wang et al., 2015c).

Seasonal measurements of $b_{sp}$ and $b_{ap}$ were also made at remote sites (Dunhuang, Yulin, and Zhangye of Gansu province, Dongsheng of Inner Mongolia) focusing on dust aerosols and only covered spring and winter (Li et al., 2010; Xu et al., 2004; Yan, 2007). $b_{sp}$ in winter ranged from 303 to 304 $Mm^{-1}$, which doubled those in spring (126 to 183 $Mm^{-1}$).

## 4. Relationships between aerosol optical properties and $PM_{2.5}$ mass concentrations

### 4.1 Mass scattering efficiency of $PM_{2.5}$

$b_{sp}$ and $PM_{2.5}$ mass concentration have been found to correlate well in numerous field studies (Andreae et al., 2008; Han et al., 2015; Hand and Malm, 2007b; Jung et al., 2009; Pu et al., 2015; Tao et al., 2014b; Tao et al., 2014c; Tao et al., 2015a; Tao et al., 2016a; Tian et al., 2015; Wang et al., 2012b; Zhao et al., 2011). A parameter

describing their relationship is defined as mass scattering efficiency (MSE), which is
the slope of the linear regression of $b_{sp}$ against $PM_{2.5}$ mass. MSE was found to vary
with location and season due to the variations in $PM_{2.5}$ chemical composition. Some
of the variations may due to different sampling conditions, e.g., ambient (controlled
RH<60%) versus dry condition (controlled RH<40%), online versus filter-based
$PM_{2.5}$ sampling. Available MSE data are discussed here, although uncertainties from
measurements will not be addressed in this study.

In BTH, annual average $PM_{2.5}$ MSE was higher in Beijing (5.9 $m^2$ $g^{-1}$) than in

rural Beijing (4.8 $m^2$ $g^{-1}$) based on online $PM_{2.5}$ mass (Zhao et al., 2011). In urban
Beijing in winter of 2013, $PM_{2.5}$ MSE increased to 4.9 $m^2$ $g^{-1}$ during a heavy
pollution episode and decreased to 3.6 $m^2$ $g^{-1}$ during clean days, due to a large
fraction of soluble inorganic components (e.g. $(NH_4)_2SO_4$ and $NH_4NO_3$) in $PM_{2.5}$
under heavy polluted conditions (Tao et al., 2015a). In rural Beijing in 2005-2010,
dust episodes had lower $PM_{2.5}$ MSE (0.7 $m^2$ $g^{-1}$) and anthropogenic pollution
episodes had higher $PM_{2.5}$ MSE (4.3 $m^2$ $g^{-1}$) (Pu et al., 2015).

In YRD, annual average $PM_{2.5}$ MSE ranged from 3.8 $m^2$ $g^{-1}$ in Ningbo to 5.3 $m^2$

$g^{-1}$ in Hangzhou with a regional urban average (including cities of Nanjing, Shanghai,
Suzhou, Hangzhou and Ningbo) of 4.1 $m^2$ $g^{-1}$ in 2011-2012 (Cheng et al., 2013b).
$PM_{2.5}$ MSE in Lin'an (4.0 $m^2$ $g^{-1}$), a rural site of YRD, was close to the regional
urban average value in YRD (Xu et al., 2002). $PM_{2.5}$ MSE in Shanghai reached 5.6
$m^2$ $g^{-1}$ in winter of 2012 (Han et al., 2015), which was higher than that in Beijing in
the same season (Tao et al., 2015a).

In PRD, annual average $PM_{2.5}$ MSE in Guangzhou was 3.5 $m^2$ $g^{-1}$ with seasonal

average ranging from 2.3 $m^2$ $g^{-1}$ in summer to 4.5 $m^2$ $g^{-1}$ in autumn in 2009-2010

(Tao et al., 2014c). These values were close to 4.2 $m^2$ $g^{-1}$ (Andreae et al., 2008) and

2.7 $m^2$ $g^{-1}$ (Jung et al., 2009) measured in the same city in autumn of 2004. However,

$PM_{2.5}$ MSE in rural Guangzhou (Wanqingsha, south of Guangzhou) was 5.3 $m^2$ $g^{-1}$

(Wang et al., 2012), which was evidently higher than that in Guangzhou in the same

season (Tao et al., 2014c).

In southwest China, seasonal average $PM_{2.5}$ MSE ranged from 3.5 to 4.4 $m^2$ $g^{-1}$

in Chengdu in 2011 (Tao et al., 2014b). In northwest China, $PM_{2.5}$ MSE was 3.0 $m^2$

$g^{-1}$ for anthropogenic pollution and 1.0 $m^2$ $g^{-1}$ for dust pollution at a remote site

(Yulin, located at the interface of the desert and loess regions, Shanxi province),

which was similar to rural Beijing (Xu et al., 2004).

In summary, annual $PM_{2.5}$ MSE typically ranged from 3.5 to 5.9 $m^2$ $g^{-1}$ in urban

areas in China with higher values in north China and lower values in south China.

Seasonal average $PM_{2.5}$ MSE typically ranged from 2.3 to 5.6 $m^2$ $g^{-1}$ with higher

values in winter and autumn and lower values in spring and summer. Generally,

$PM_{2.5}$ MSE typically ranged from 3.0 to 5.0 $m^2$ $g^{-1}$ for anthropogenic pollution and

from 0.7 to 1.0 $m^2$ $g^{-1}$ for natural dust aerosols.

**4.2 Mass absorption efficiency of EC and organic matter**

EC is the dominant absorption species in $PM_{2.5}$. Similar to $PM_{2.5}$ MSE, the slope

between $b_{ap}$ and EC mass was defined as mass absorption efficiency (MAE) of EC.

Various instruments have been used to measure $b_{ap}$ including aethalometer,

multi-angle absorption photometer (MAAP), Radiance Research Particle Soot
Absorption Photometer (PSAP), and Photoacoustic Spectrometer (PAS), with the
former two instruments measuring attenuation of the sample on the filter for
estimating BC mass concentration, and the latter two measuring $b_{ap}$ directly. Most
studies in China used an aethalometer and MAAP. BC mass concentrations (880nm)
were converted to $b_{ap}$ (532nm) using an empirical constant of 8.28 $m^2 g^{-1}$, which was
obtained by the regression between BC mass and $b_{ap}$ synchronously measured in
autumn in Guangzhou, keeping in mind that application of an empirical constant
obtained from one specific study to other cases may cause large uncertainties (Wu et
al., 2009).
EC MAE was 7.5-8.5 $m^2 g^{-1}$ in winter and 9.4 $m^2 g^{-1}$ in summer in Beijing (632
nm) (Cheng et al., 2011; Wu et al., 2016). The higher EC MAE in summer was likely
due to more coating of EC in higher ambient humidity (Wu et al., 2016). BC MAE
was 6.5 $m^2 g^{-1}$ at 532 nm in autumn in Shenzhen of PRD (Lan et al., 2013). However,
BC MAE was 12.4 $m^2 g^{-1}$ at 532 nm in winter in Xi'an (Wang et al., 2014a).
Moreover, EC MAE of diesel was 8.4 $m^2 g^{-1}$ (632 nm), which was higher than those
(3.0-6.8 $m^2 g^{-1}$) of biomass burning sources (e.g. crop residual and wood) (Cheng et
al., 2011).
Organic matter or brown carbon is also a strong light absorbing material at short
wavelengths. Available MAE values of OC include 0.76 $m^2 g^{-1}$ (532 nm) in autumn
in 2008 in Guangzhou (Andreae et al., 2008). Moreover, available MAE values of
WSOC include 1.79 and 0.71 $m^2 g^{-1}$ (365nm) in winter and summer, respectively, in
Beijing (Cheng et al., 2011). The WSOC MAEs of wood, grass, corn, and diesel
tractors were 0.97, 0.90, 1.05, and 1.33 $m^2$ $g^{-1}$ (365nm), respectively, which were
much higher than that of gasoline motorcycles (0.20 $m^2$ $g^{-1}$, 365nm) (Du et al.,
2014b). Evidently, the MAEs of OC or WSOC should not be neglected for short
wavelength absorption.

## 1069 4.3 Aerosol hygroscopic properties

$b_{sp}$ under ambient condition can differ significantly from dry conditions due to
hygroscopic properties of soluble aerosol chemical components. A relationship
between ambient and dry $b_{sp}$ is thus developed for estimating ambient $b_{sp}$ from
measured dry $b_{sp}$, which is often described by the hygroscopic growth curve ($f$(RH))
as a function of RH: $f(RH)=1+a\times(RH/100)^b$. Here, $a$ and $b$ are empirical fitting
parameters. Only a few studies conducted in Beijing, Wuqing, Lin'an and
Guangzhou provided the aerosol hygroscopic curves (Table S4 of the supplement
document). Three different methods have been used to obtain $f$(RH). The first one
measures simultaneously dry and wet $b_{sp}$ using a nephelometer and visibility meter,
respectively. The second one measures wet $b_{sp}$ by integrating a nephelometer
equipped with a humidifier, and the third one estimates dry and wet $b_{sp}$ based on Mie
theory with size-resolved chemical components.
Available $f$(RH) curves in China are summarized in Fig. 14. The three $f$(RH)
curves in autumn of 2007, 2011 and 2014 in urban Beijing were all measured using
the first method (Fig.14 a) (Liu et al., 2013b; Liu et al., 2013c; Yang et al., 2015).
The two $f$(RH) curves measured in 2011 and 2014 were quite close, but the one in
2007 was lower under RH< 80% and higher under RH>80%, likely due to aerosol
chemical composition and size distribution changes in these years.

The $f$(RH) curves at four rural sites were measured using the second method,

including Baodi of Tianjin in spring, Wuqing of Tianjin in winter (Fig.14 b) (Chen et
al., 2014b; Pan et al., 2009), Raoyang of Hebei province in summer (Wu et al., 2017),
and Lin'an of Zhejiang province in spring (Fig.14 c) (Zhang et al., 2015b). It is
known that the hygroscopic chemical components are mostly water-soluble inorganic
salts (e.g. $(NH_4)_2SO_4$, $NH_4NO_3$), while mineral dust and organic matter are mostly
hydrophobic. In Baodi in spring season, the concentrations of $(NH_4)_2SO_4$ and
$NH_4NO_3$ and their mass fractions in fine mode particles were higher during polluted
episodes than during clean periods or dust storm episodes, resulting in higher $f$(RH)
values during the polluted episode. $f$(RH) values measured in winter in Wuqing were
evidently higher than those measured in spring in Baodi under RH<80% likely due
to more hygroscopic chemical components in winter in Wuqing. In Raoyang, a
different fitting curve of $f$(RH) was obtained with a much higher $f$(RH=80%) value
(2.3) than in other rural sites in BTH mentioned above, likely due to higher fractions
of hydrophilic components in $PM_{2.5}$ (>56%). In all the BTH sites, $f$(RH) value
increased continuously with increasing RH. However, in a different study an abrupt
increase in $f$(RH) at RH values of 73-81% was observed in summer in Wangdu of
Hebei province due to the deliquescence of ammonium sulfate (Kuang et al., 2016).
Similar to what was found in Baodi, $f$(RH) values during polluted episodes were also
higher than those during dust episodes in Lin'an, but the differences between
polluted and dust periods were smaller in Lin'an than in Baodi. Noticeably, the $f$(RH)
values during polluted episodes were similar in Lin'an and Baodi, e.g. $f$(RH=80%)
was 1.5 and 1.6, respectively.
The $f$(RH) curves (solid lines) in summer in urban Guangzhou were measured by
the first method, while those (dot lines) in autumn in rural Guangzhou and in
summer and autumn seasons in urban Guangzhou were measured by the third
method (Fig.14 d) (Cheng et al., 2008b; Lin et al., 2014; Liu et al., 2008a).
$f$(RH=80%) values were 2.04 and 2.68, respectively, for urban aerosols originating
from the north and marine aerosols originating from the South China Sea. $f$(RH<80%)
curves were similar in urban and rural Guangzhou; however, $f$(80%<RH<90%)
values in rural Guangzhou were evidently higher than those in urban Guangzhou,
likely due to the much higher fraction of secondary inorganic aerosols in fine mode
particles in rural Guangzhou than urban Guangzhou in the dry season (Lin et al.,
2014; Liu et al., 2008b).
If averaging all available $f$(RH) curves shown in Figure 15, the empirical fitting
parameters $a$ and $b$ were found to be 2.87±0.03, 5.50±0.06, respectively (Fig 15a).
But if excluding dust episodes in Baodi and Lin'an (Fig 15 b), the empirical fitting
parameters $a$ and $b$ were 3.17±0.03, 5.54±0.06, respectively (Figure 15b). Based on
the average $f$(RH) curve, $b_{sp}$ under ambient condition (RH=80%) can be amplified by
about 1.8 times of that under dry conditions in China. This suggests that reducing
inorganic water-soluble salts is critical in alleviating hazy weather in China.

## 4.4 Source apportionment of haze in China


To investigate the contributions of $PM_{2.5}$ chemical components to $b_{sp}$, a revised
formula developed by the original IMPROVE method is applied in this section
(Pitchford et al., 2007). The revised IMPROVE formula can be simplified as follows:
$b_{ext} \approx 2.2 \times f_S \times$ [Small $(NH_4)_2SO_4$] + 4.8 $\times f_L \times$ [Large $(NH_4)_2SO_4$] + 2.4 $\times f_S \times$
[Small $NH_4NO_3$] + 5.1 $\times f_L \times$ [Large $NH_4NO_3$] + 2.8 $\times$ [Small OM] + 6.1 $\times$ [Large
OM] + 1.0 $\times$ [Other]+ 10 $\times$ [EC]                                      (1)
[Large X] = [Total X]$^2$ / 20, for [Total X] < 20                         (2)
[Large X] = [Total X], for [Total X] $\geq$ 20                             (3)
[Small X] = [Total X] - [Large X]                                          (4)
Where X represents $(NH_4)_2SO_4$, $NH_4NO_3$ and OM, respectively. RH growth
curves of $f_S$ and $f_L$ for $(NH_4)_2SO_4$ and $NH_4NO_3$ can be found in Pitchford et al.

(2007).

Using the chemical composition data shown in Fig. 8 and annual average RH
values in major cities in China as input (http://data.cma.cn/), the estimated annual $b_{ext}$
and its load percentages under dry and ambient conditions are plotted in Fig. 16. For
$b_{ext}$ under dry conditions, carbonaceous aerosols had similar percentage contributions
as secondary inorganic aerosols in Shijiazhuang, Tianjin, Shangdianzi, Shanghai, Hok
Tsui, Zhengzhou, Xi'an, Jinan, Chengdu, Fuzhou and Xiamen, but the percentage
contributions were 11-65% higher in other urban and rural sites. However, under
ambient conditions the contributions of secondary inorganic aerosols were evidently
higher (by 2-54%) than those of carbonaceous aerosols in most cities except in
Beijing, Chengde, Lanzhou and Chongqing. Noticeably, the contributions of
secondary inorganic aerosols for $b_{ext}$ sharply increased by about 18-25% under
ambient conditions than dry conditions in humid (RH>70%) cities (e.g. Haikou,
Changsha, Xiamen, Nanjing, cities in PRD, and Chengdu).

## 5. Implications for aerosol pollution controls

There is no doubt that reduction of $PM_{2.5}$ will be the ultimate approach for
improving visibility and alleviating hazy weather. Industrial emission contributions
to secondary inorganic aerosols were the dominant sources of $PM_{2.5}$ in urban areas in
China (Liang et al., 2016). Aerosols produced from traffic emissions, biomass
burning and soil dust were also important sources in north China. Secondary
inorganic aerosols were formed from atmospheric reactions involving $SO_2$ and $NO_x$,
which were mainly emitted from coal combustion, the major energy source in China
for decades.
A series of regulations controlling coal combustion emissions has been made
since the first version of the NAAQS was promulgated in 1982. The Air Pollution
Prevention law of PRC was promulgated in 1987, which was the milestone in air
pollution prevention history in China. It also marked the beginning of a new era for
preventing air pollution based on the national law, followed by a series of regulations
for controlling coal combustion. During 1990-2000, most of the control measures or
technologies (e.g., desulfurization and dedusting for coal combustion) were focused
on reducing $SO_2$ emissions. The measure for gross control of $SO_2$ emissions was
enforced since 1996. Despite these efforts, the amount of $SO_2$ emissions increased
by about 28% from 2000 to 2005 (http://www.zhb.gov.cn/). The amount of $SO_2$
emission began to decrease in 2006 and gradually reduced to the emission level of
2000   in   2010   (http://www.zhb.gov.cn/).   Meanwhile,   ambient   annual   $SO_2$
concentration in urban cities in China also decreased from 57 $\mu g\ m^{-3}$ in 2005 to 40
$\mu g\ m^{-3}$ in 2010 (http://www.zhb.gov.cn/). Apparently, the emission control efforts for
reducing $SO_2$ emissions since 2006 have been effective.
The control measures for $NO_2$ only began with the control of vehicular emissions
in 1995, but the inclusion of $NO_2$ in the gross control indexes did not happen until
2010. New coal power plants were also required to denitrate after 2010. The
emissions of $NO_x$ actually increased from 1996 to 2010, as seen in the vertical
column $NO_2$ derived from satellite data (Zhang et al., 2012b). Although annual
average ambient $NO_2$ at the surface fluctuated from 30-40 $\mu g\ m^{-3}$ during 2000-2010
in China (http://www.zhb.gov.cn/), annual average ambient $NO_2$ in megacities (e.g.
Beijing, Shanghai and Guangzhou) slowly increased. Evidently, the control of
emissions of nitrate gaseous precursors was not very effective during 2000-2010.
Despite the above-mentioned control measures, sulfate concentrations remained
high and nitrate concentrations even gradually increased in megacities in China.
More recently, the Clean Air Action Plan (CAAP) for improving air quality was
promulgated and implemented by the State Council of the People's Republic of
China in 2013 (http://www.gov.cn). This plan aims to reduce the $PM_{2.5}$ annual mass
concentrations by 25%, 20%, and 15% of the 2012 levels in BTH, YRD, and PRD,
respectively. The key industries including power plants, iron and steel smelting,
petroleum chemical, cement, nonferrous metals smelting, and chemical production
were required to execute stricter emission standards in the key regions including
most megacities in China (http://www.zhb.gov.cn). Accordingly, annual average
$PM_{2.5}$ in China from online monitored data at 74 cities gradually decreased from 72
$\mu g\ m^{-3}$ in 2013 to 50 $\mu g\ m^{-3}$ in 2015, showing some promising results from the series
of control measures.
One factor that needs to be considered in future pollution reduction is the
non-linearity of chemistry (Cheng et al., 2016). For example, a model sensitivity
study suggested potential increase in $NO_3^-$ mass concentrations due to the increased
atmospheric oxidizing capacity, even with decreasing $NO_x$ emissions (Zhao et al.,
2013a). Furthermore, increased atmospheric oxidizing capacity may also enhance the
conversion of VOCs to OM. In fact, the contribution of secondary organic aerosols
to $PM_{2.5}$ was also high and could increase further in typical megacities in China (He
et al., 2011; Huang et al., 2014b; Sun et al., 2013). Another factor that requires more
attention is ammonia emissions from agricultural activities in rural areas and human
activities in cities. Ammonia emissions can enhance $PM_{2.5}$ pollution substantially,
especially in ammonia-limited (acid aerosols) areas (Wang et al., 2011). This topic
needs further investigation through both modeling simulation and field observations.
To improve the air quality across China, the following recommendations are
provided based on the major chemical components contributing to $PM_{2.5}$ and their
impact on aerosol optical properties. Emissions produced from coal combustion, in
both the industrial sectors and in residential areas, need to be further reduced. While
advanced pollution control technologies should be adopted in the medium term in
major industrial sectors consuming coal, cleaner energy sources should be
considered a long-term goal (Cao et al., 2016). Providing cleaner energy to the vast
rural and urban areas in north China for heating and cooking can not only reduce
coal combustion emissions but also biomass burning emissions. Efficient use of
fertilizers in agriculture is needed to reduce nitrogen emissions especially ammonia
(Behera et al., 2013). Educating the public to reduce meat consumption in their daily
lives, especially in the more affluent developed regions, can reduce the nitrogen
footprint substantially and thus nitrogen emissions (Galloway et al., 2014), besides
the potential benefits to human health. Traffic emissions in megacities may also need
to be constrained, such as developing more efficient public transportation systems
and limiting the use of personal automobiles. Having more vegetation coverage is
especially important in arid or semi-arid areas as well as urban areas in reducing dust
emissions (Baldauf, 2017), aside from the biological benefits. The continued
expansion of the three northern region shelter forests in north China can potentially
reduce dust emissions by increasing the dry deposition removal of aerosols (Zhang et
al., 2017b).

## 1235 Acknowledgements

This study was supported by the National Natural Science Foundation of China
(No. 41475119).

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

**List of Figures**

Fig. 1. Geographical regions and location of cities with measurements.

Fig. 2. Annual $PM_{2.5}$ mass concentration in various Chinese cities having filter-based measurements: (a) categorized into regions, and (b) lined with latitude.

Fig. 3. Inter-annual variations in $PM_{2.5}$ and dominant chemical components in Beijing (a), Shanghai (b) and Guangzhou(c).

Fig. 4. Seasonal $PM_{2.5}$ and dominant chemical components in BTH.

Fig. 5. Seasonal $PM_{2.5}$ and dominant chemical components in YRD.

Fig. 6. Seasonal $PM_{2.5}$ and dominant chemical components in PRD.

Fig. 7. Seasonal $PM_{2.5}$ and dominant chemical components in other cities.

Fig. 8. Annual $PM_{2.5}$ and dominant chemical components in China.

Fig. 9. Inter-annual variations in $PM_{2.5}$ and the dominant six sources in Beijing.

Fig. 10. Annual contributions of $PM_{2.5}$ dominant sources across China.

Fig. 11. Spatial distribution of annual average AOD across China in 2014.

Fig. 12. Annual $b_{sp}$ and $b_{ap}$ in China.

Fig. 13. Seasonal $b_{sp}$ and $b_{ap}$ in cities with measurements.

Fig. 14. The hygroscopic growth curves in different sites in China.

Fig. 15. Distribution of the hygroscopic growth curves in China.

Fig. 16. Annual $b_{ext}$ percentage loading under dry and ambient conditions at urban sites in China.

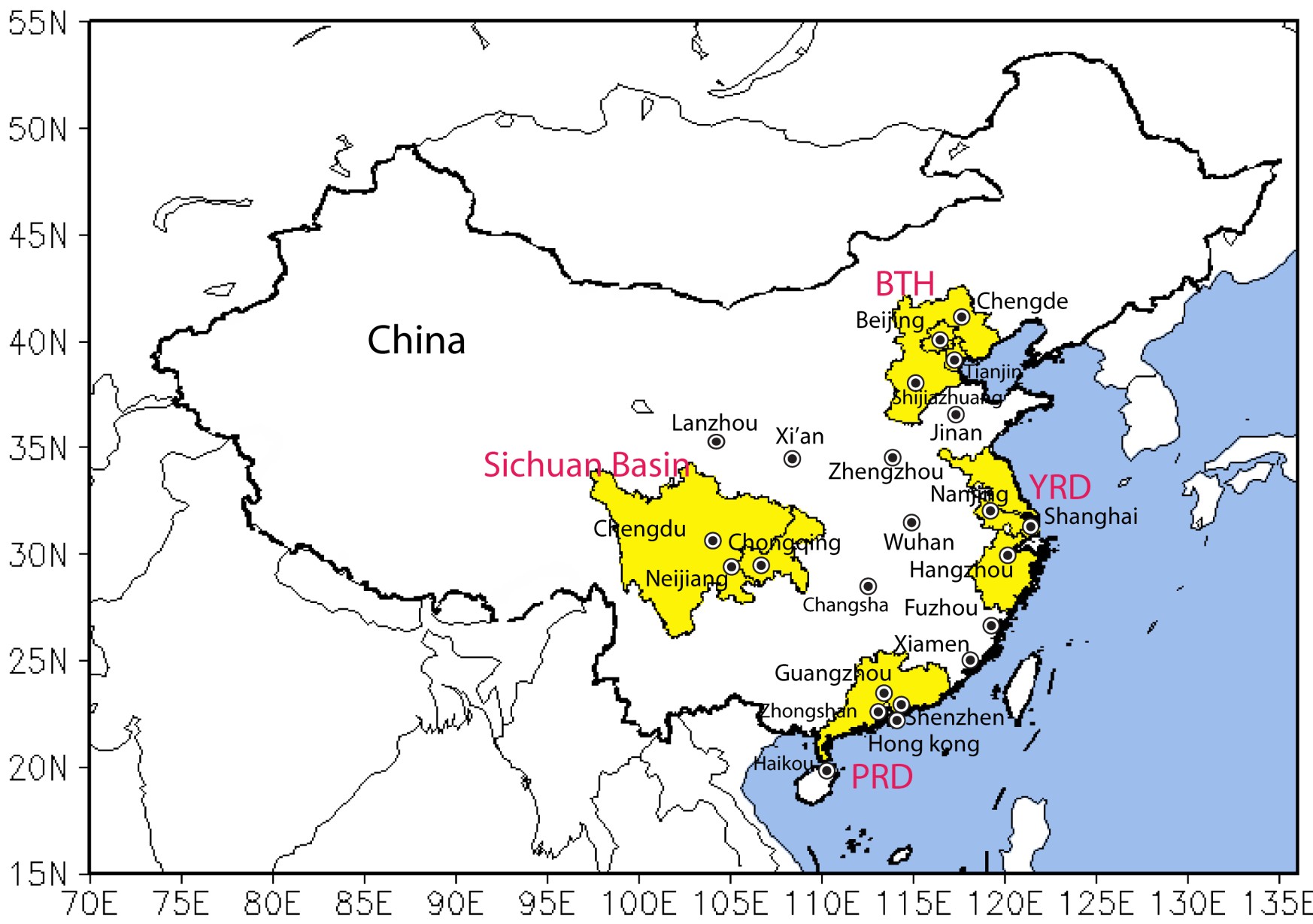

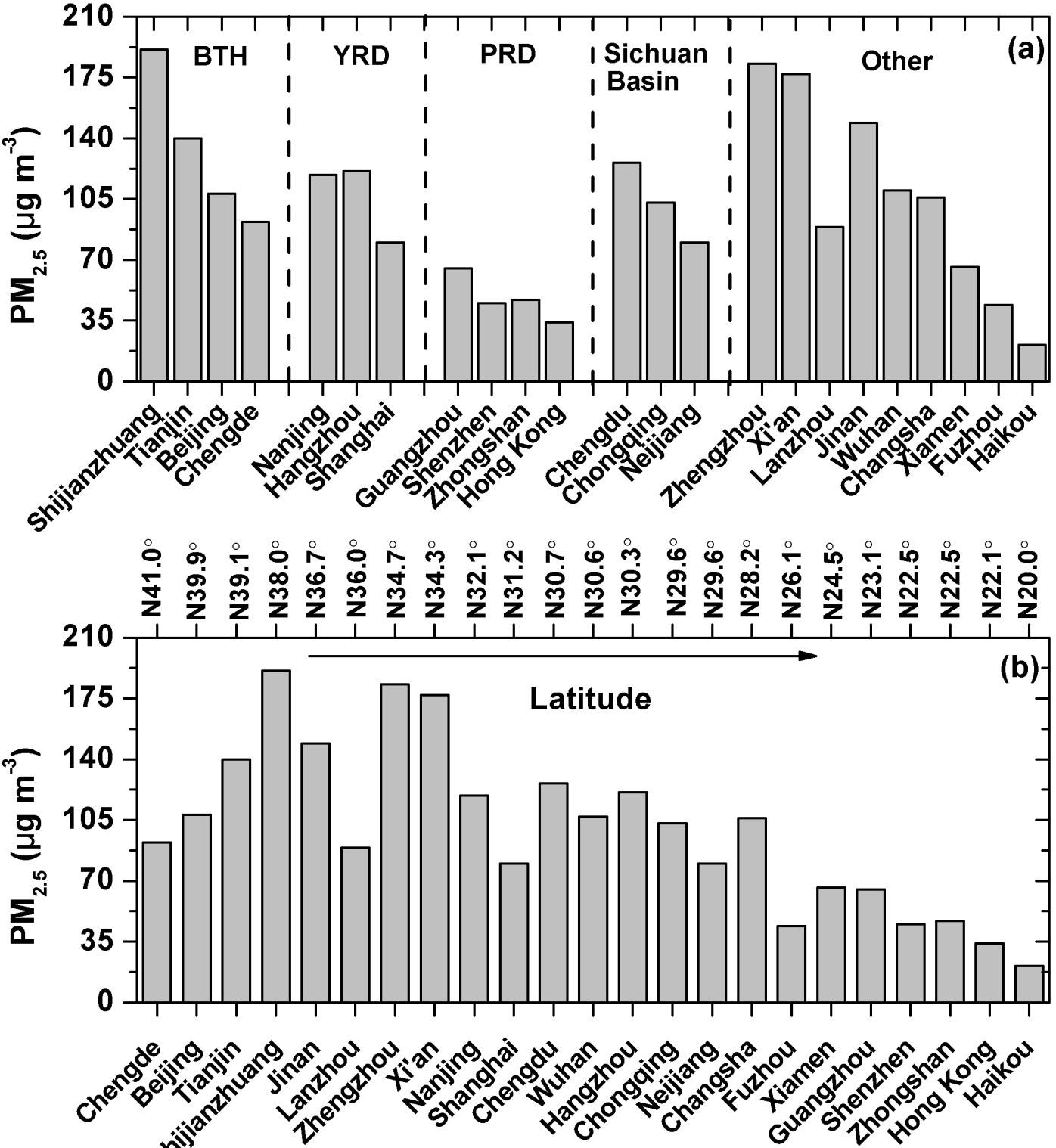

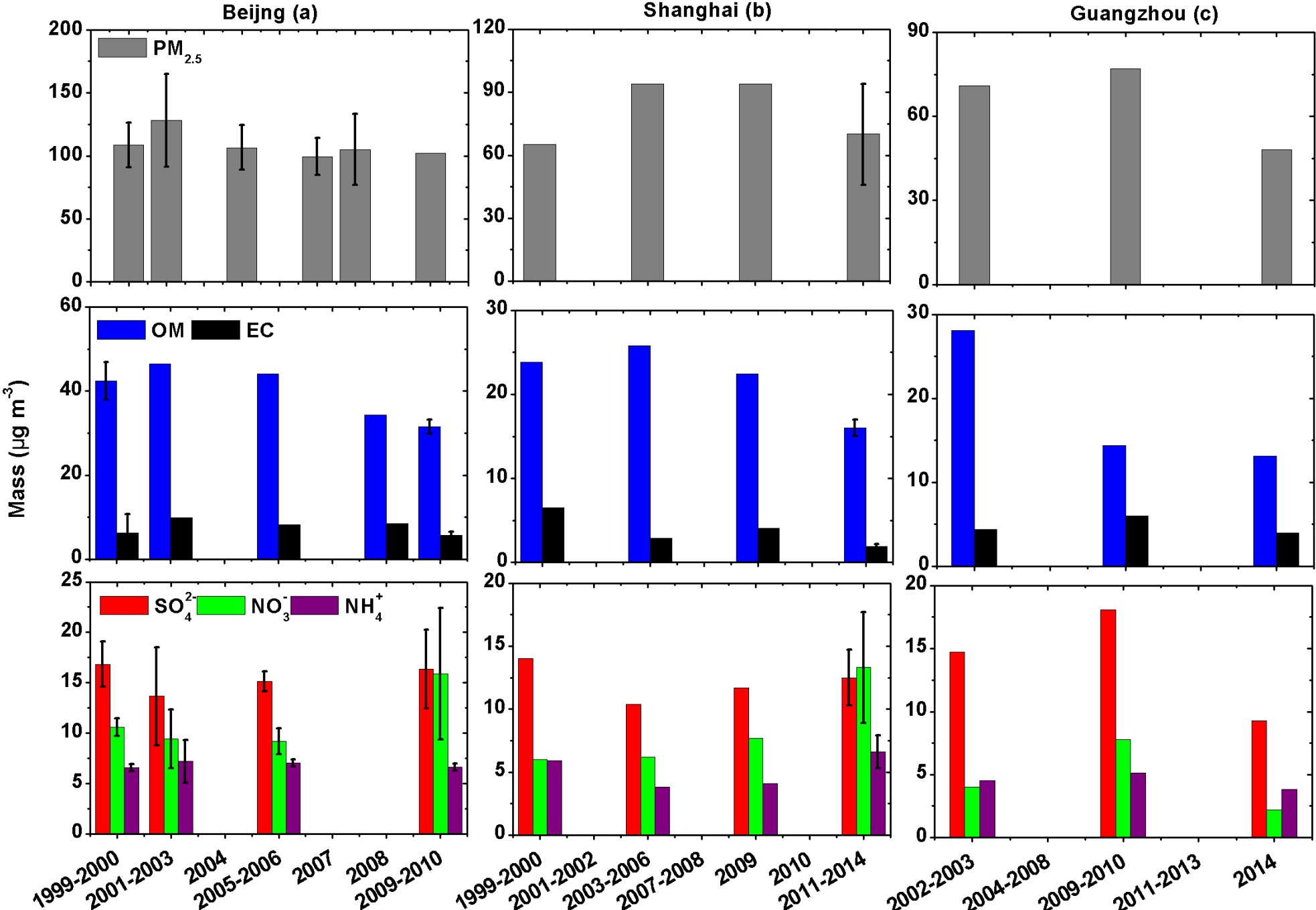

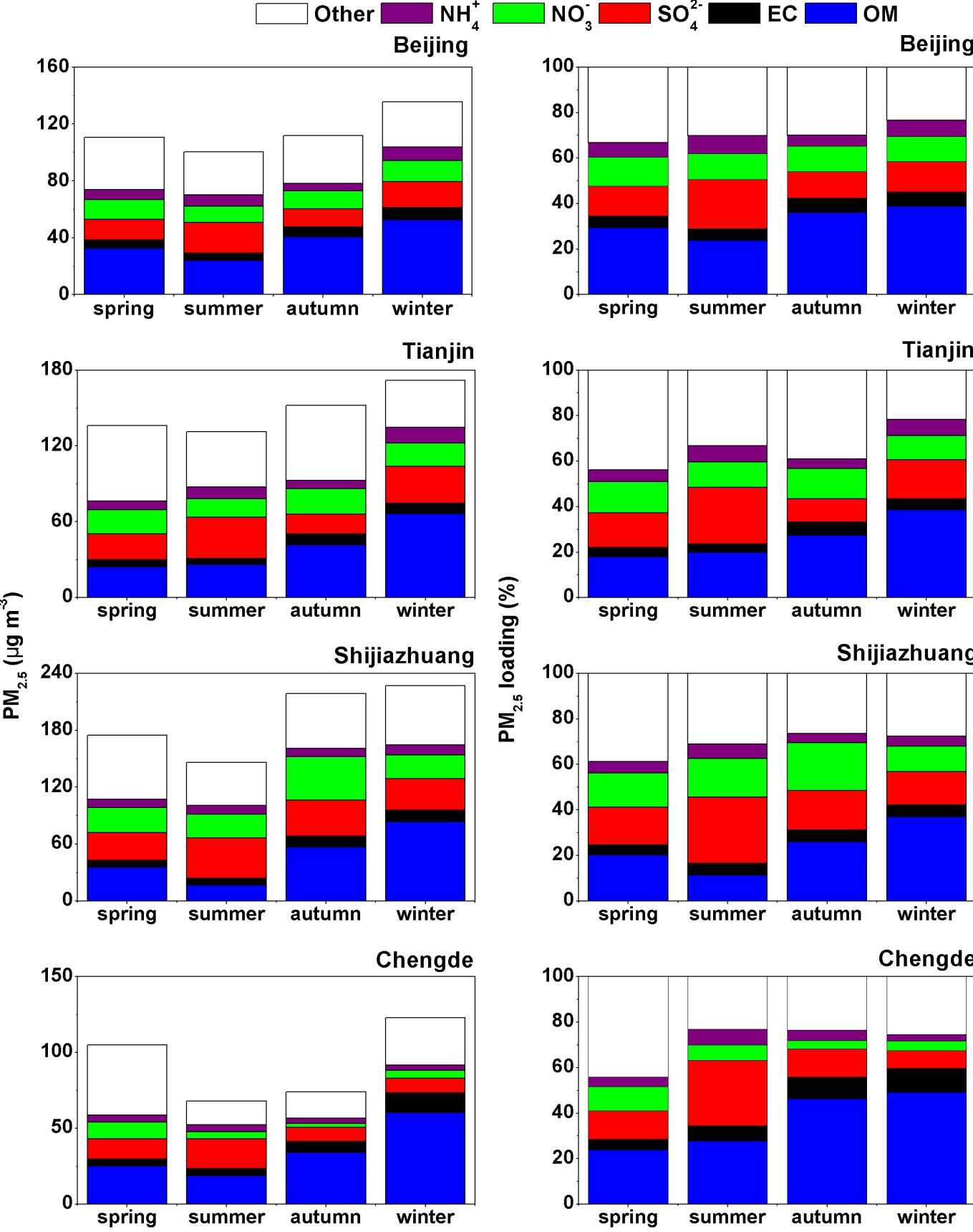

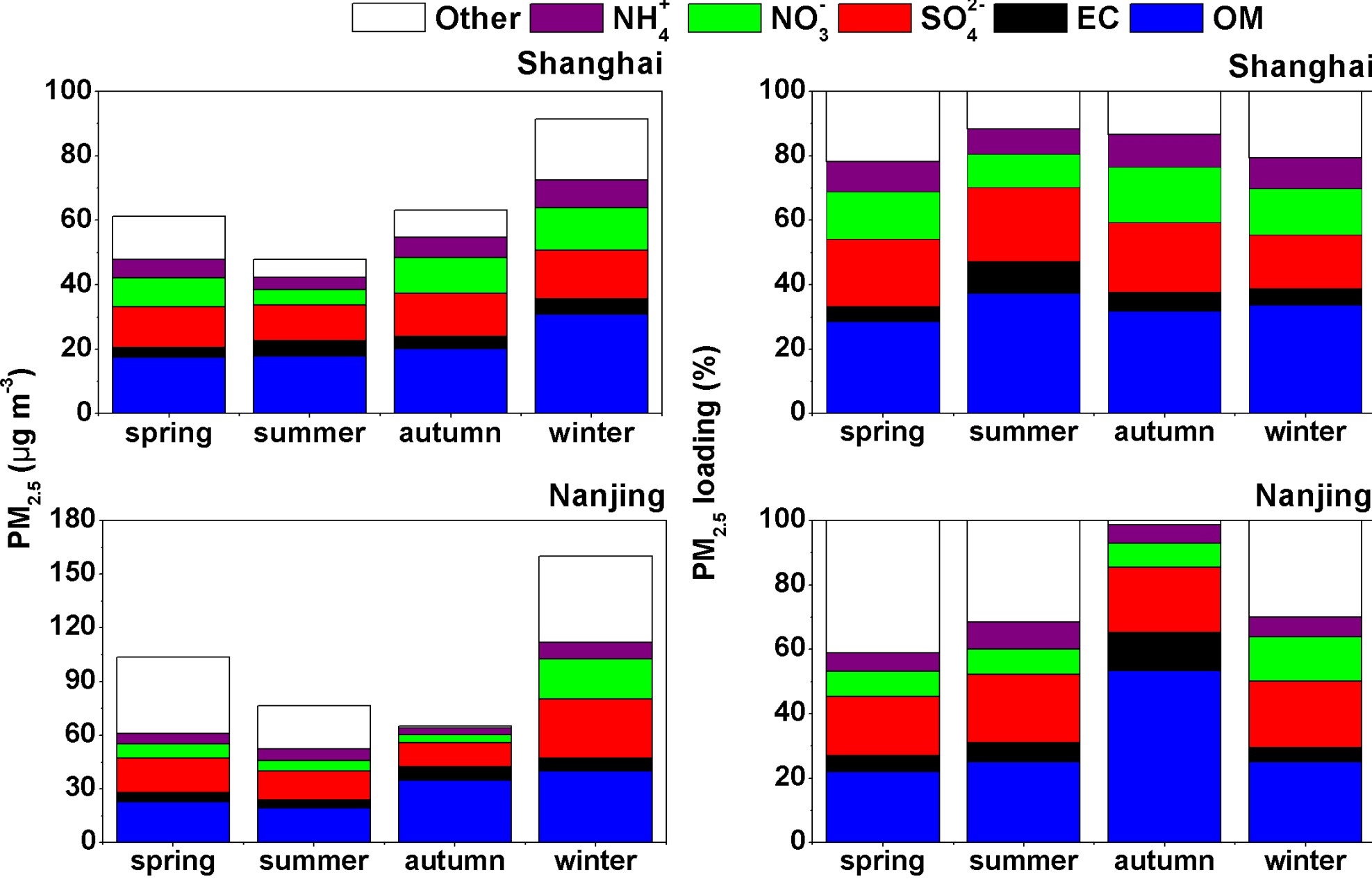

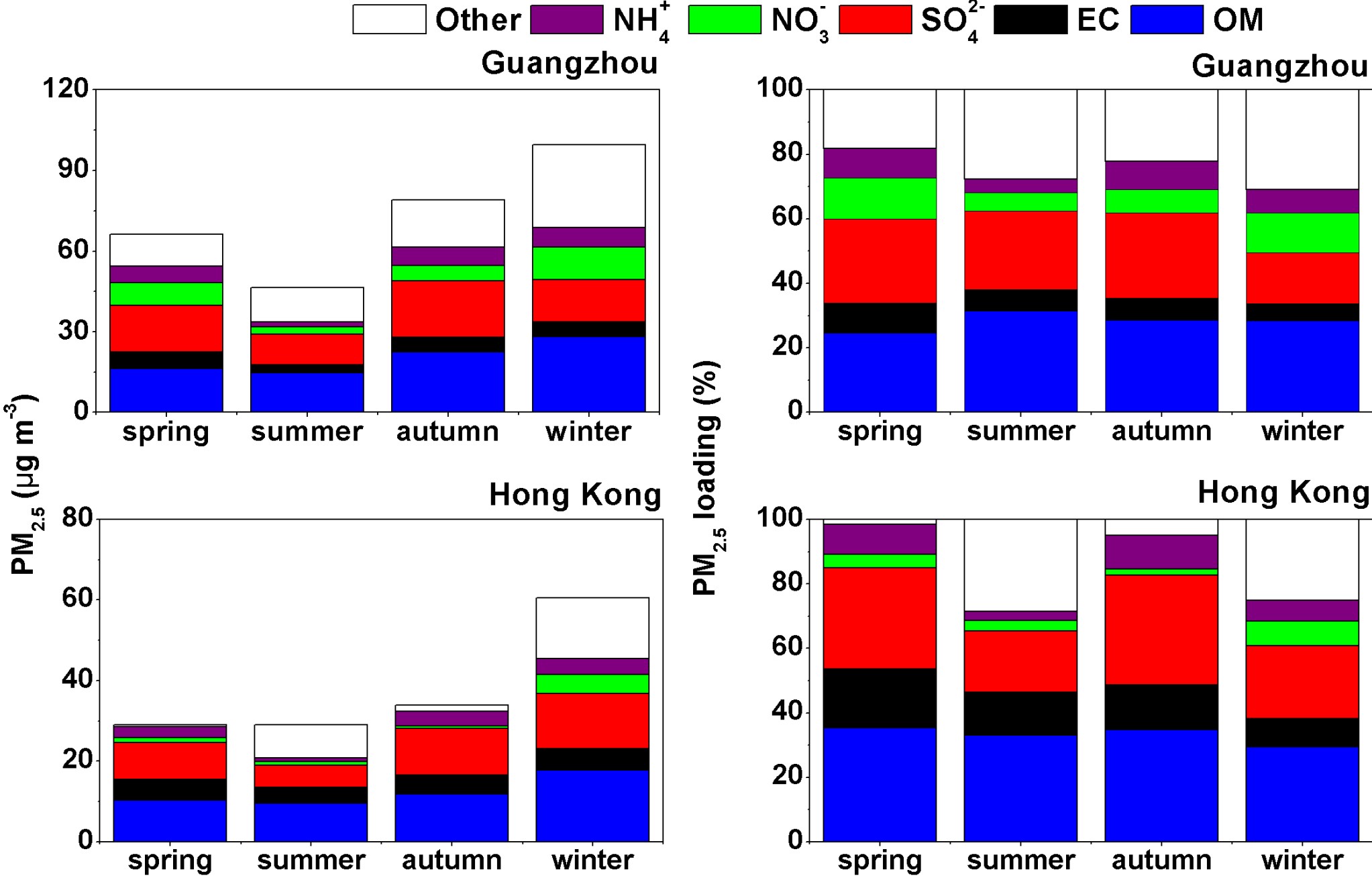

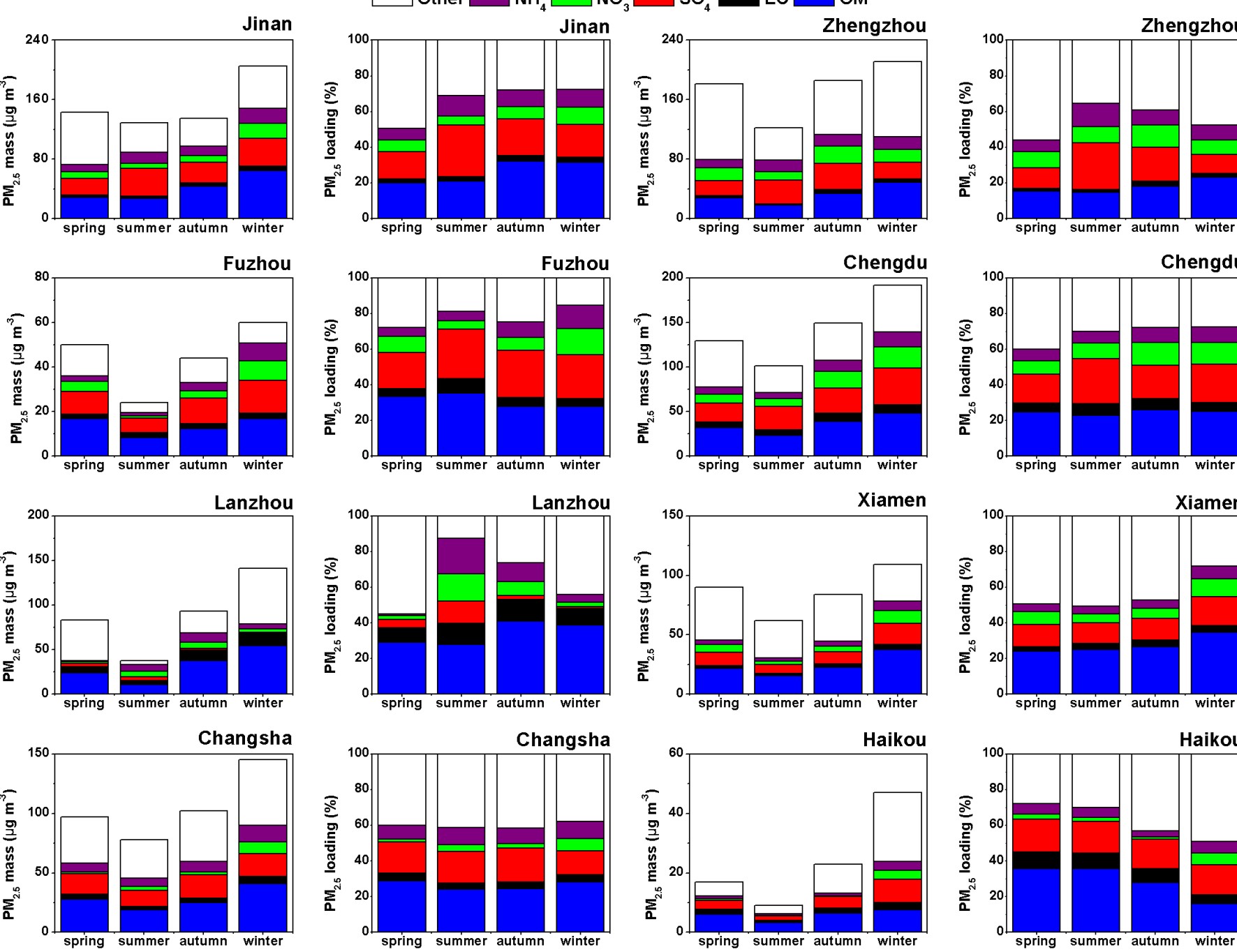

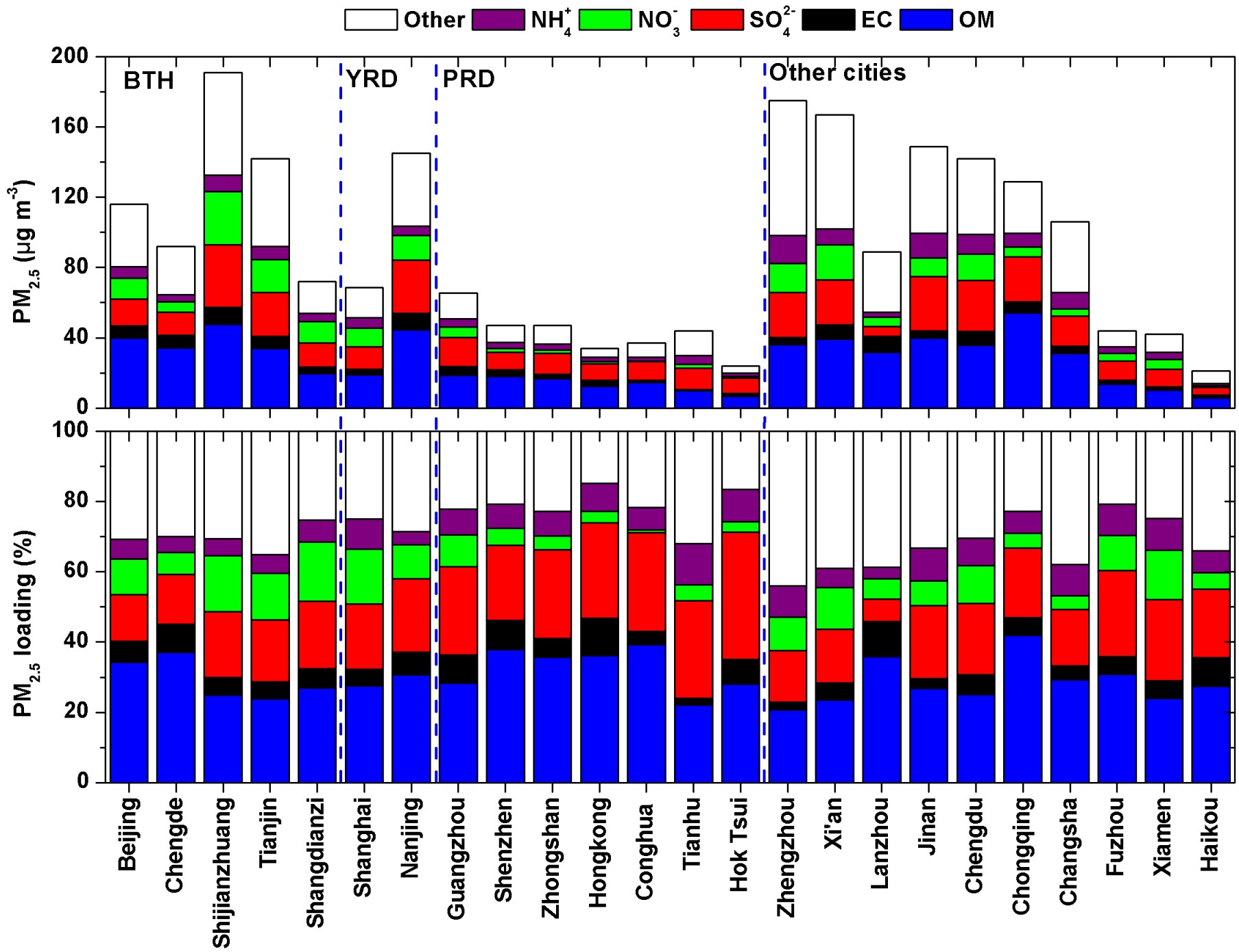

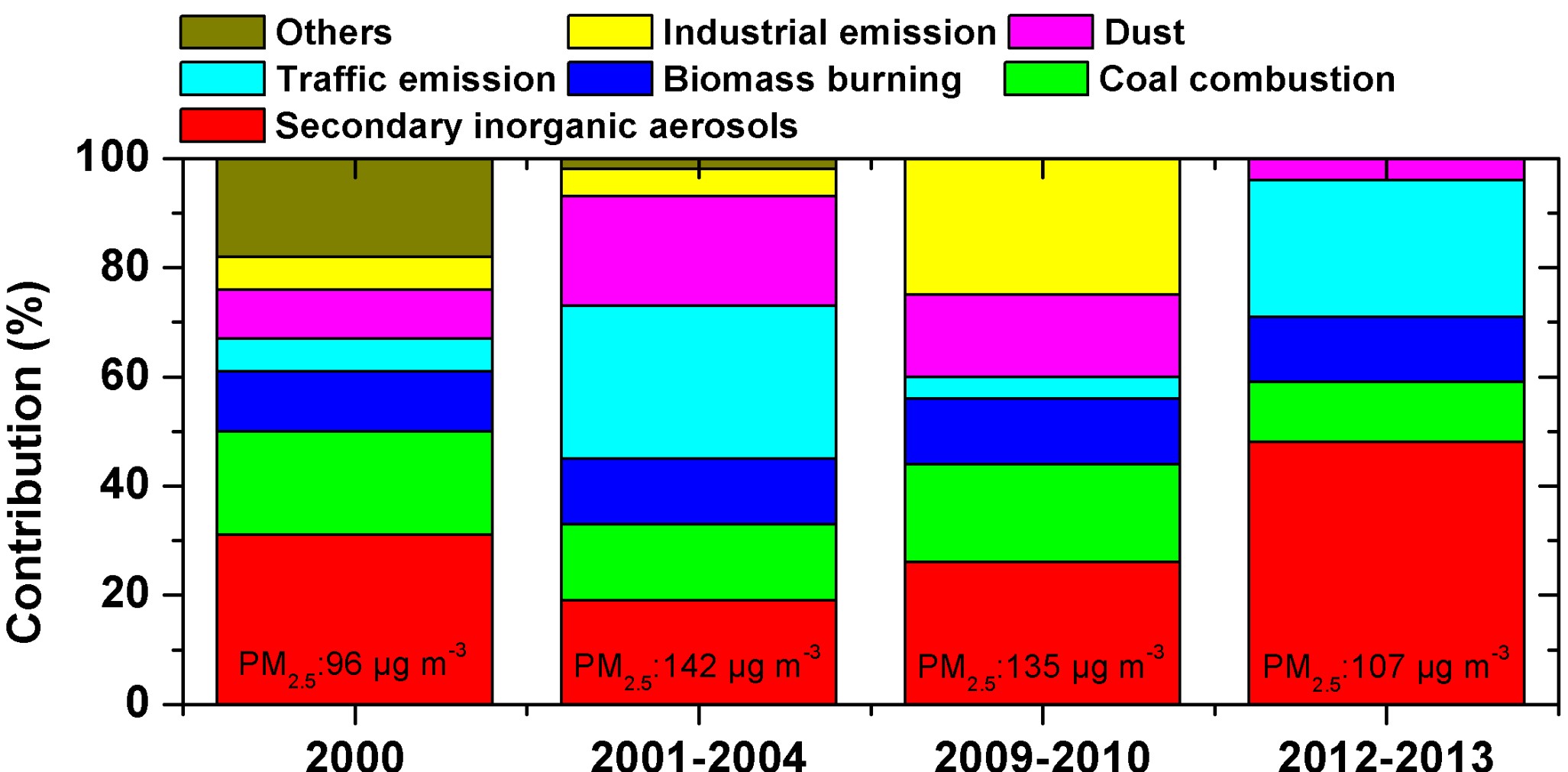

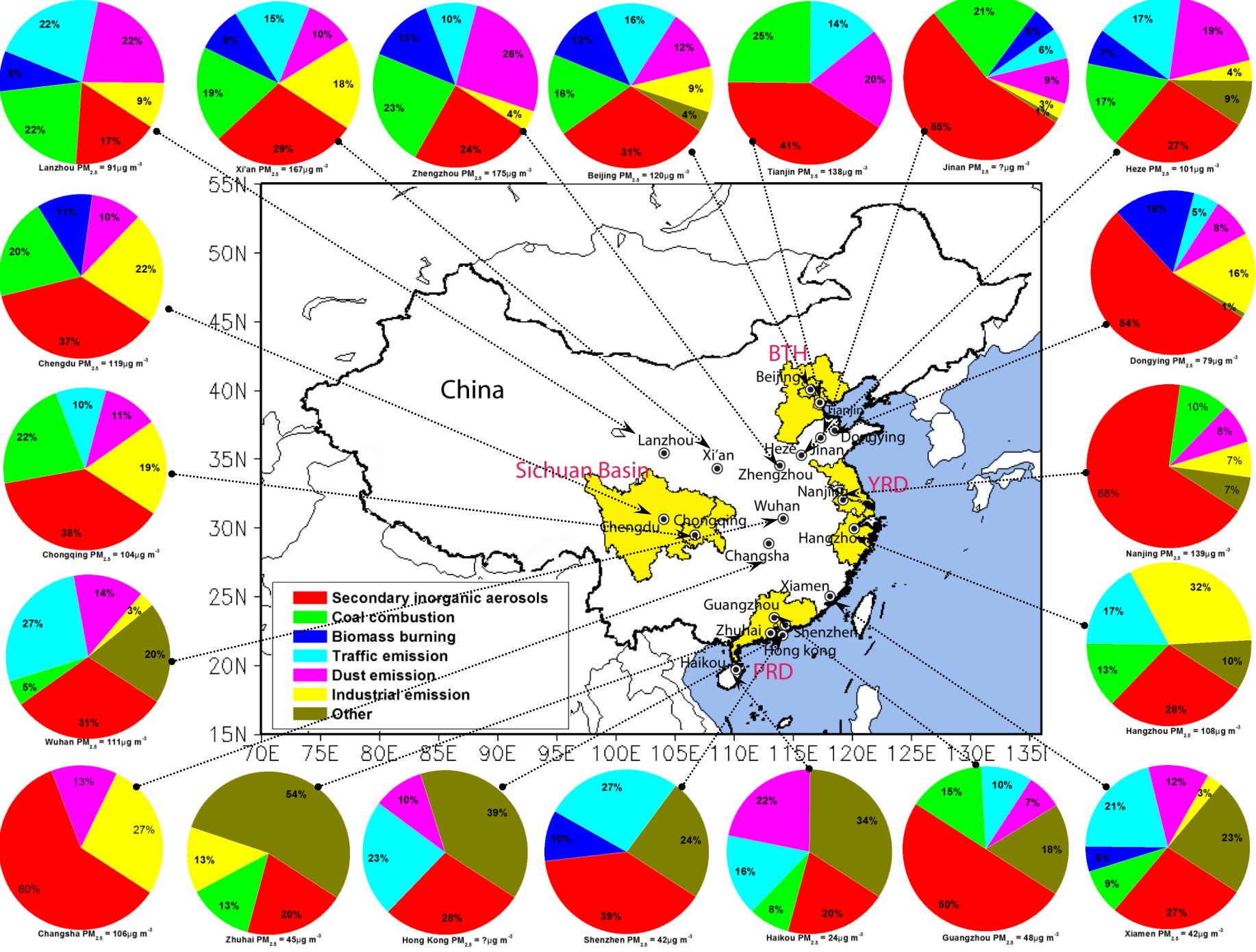

Lanzhou PM$_{2.5}$ = 91μg m$^{-3}$

Xi'an PM$_{2.5}$ = 167μg m$^{-3}$

Zhengzhou PM$_{2.5}$ = 175μg m$^{-3}$

Beijing PM$_{2.5}$ = 120μg m$^{-3}$

Tianjin PM$_{2.5}$ = 138μg m$^{-3}$

Jinan PM$_{2.5}$ = ?μg m$^{-3}$

Heze PM$_{2.5}$ = 101μg m$^{-3}$

Chengdu PM$_{2.5}$ = 119μg m$^{-3}$

Dongying PM$_{2.5}$ = 79μg m$^{-3}$

Chongqing PM$_{2.5}$ = 104μg m$^{-3}$

Nanjing PM$_{2.5}$ = 139μg m$^{-3}$

Wuhan PM$_{2.5}$ = 111μg m$^{-3}$

Hangzhou PM$_{2.5}$ = 108μg m$^{-3}$

Changsha PM$_{2.5}$ = 106μg m$^{-3}$

Zhuhai PM$_{2.5}$ = 45μg m$^{-3}$

Hong Kong PM$_{2.5}$ = ?μg m$^{-3}$

Shenzhen PM$_{2.5}$ = 42μg m$^{-3}$

Haikou PM$_{2.5}$ = 24μg m$^{-3}$

Guangzhou PM$_{2.5}$ = 48μg m$^{-3}$

Xiamen PM$_{2.5}$ = 42μg m$^{-3}$

China

BTH

Sichuan Basin

YRD

PRD

Beijing
Tianjin
Dongying
Heze Jinan
Zhengzhou
Nanjing
Lanzhou
Xi'an
Wuhan
Chengdu Chongqing
Hangzhou
Changsha
Xiamen
Guangzhou
Zhuhai Shenzhen
Haikou Hong kong

Secondary inorganic aerosols
Coal combustion
Biomass burning
Traffic emission
Dust emission
Industrial emission
Other

55N
50N
45N
40N
35N
30N
25N
20N
15N

70E 75E 80E 85E 90E 95E 100E 105E 110E 115E 120E 125E 130E 135E

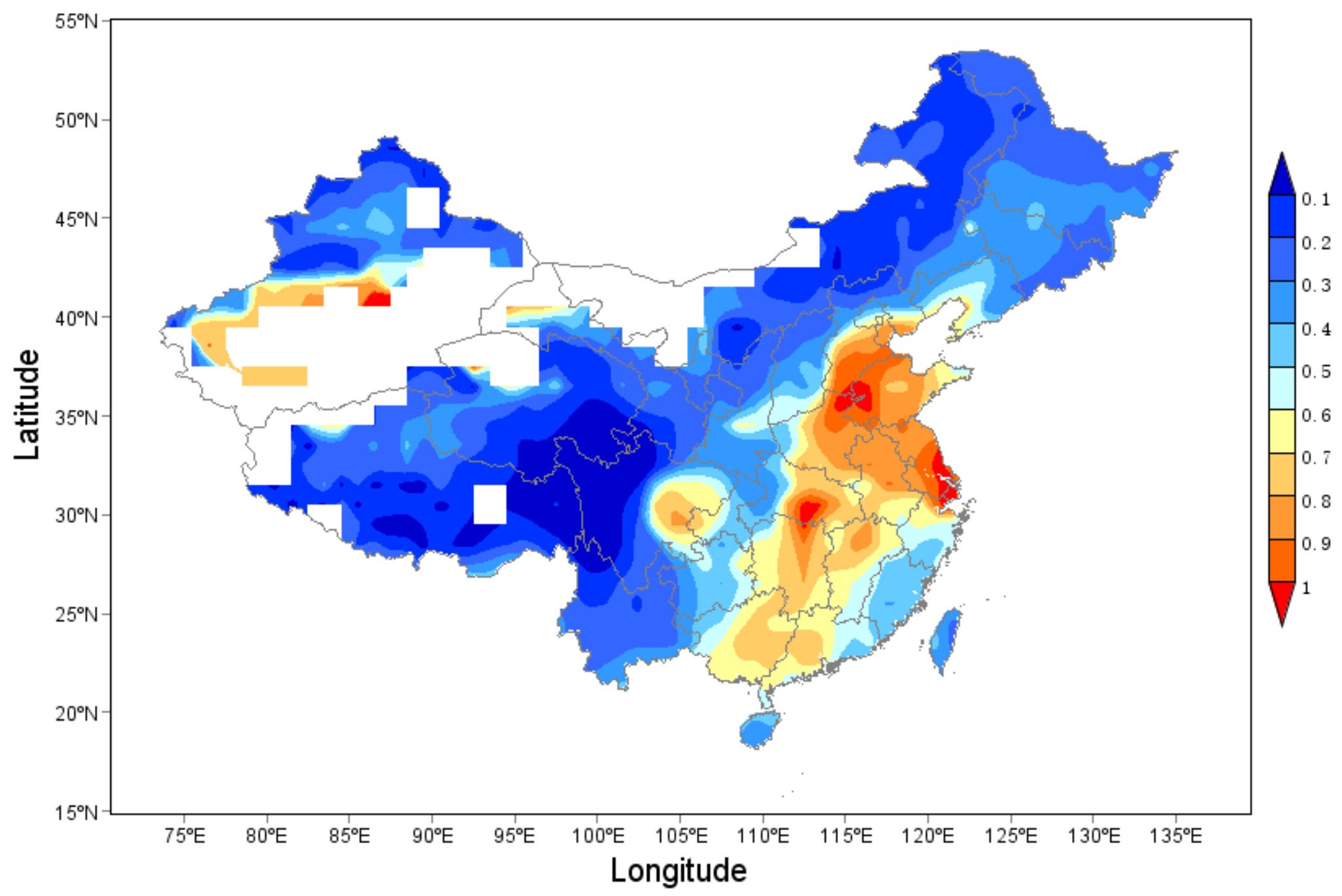

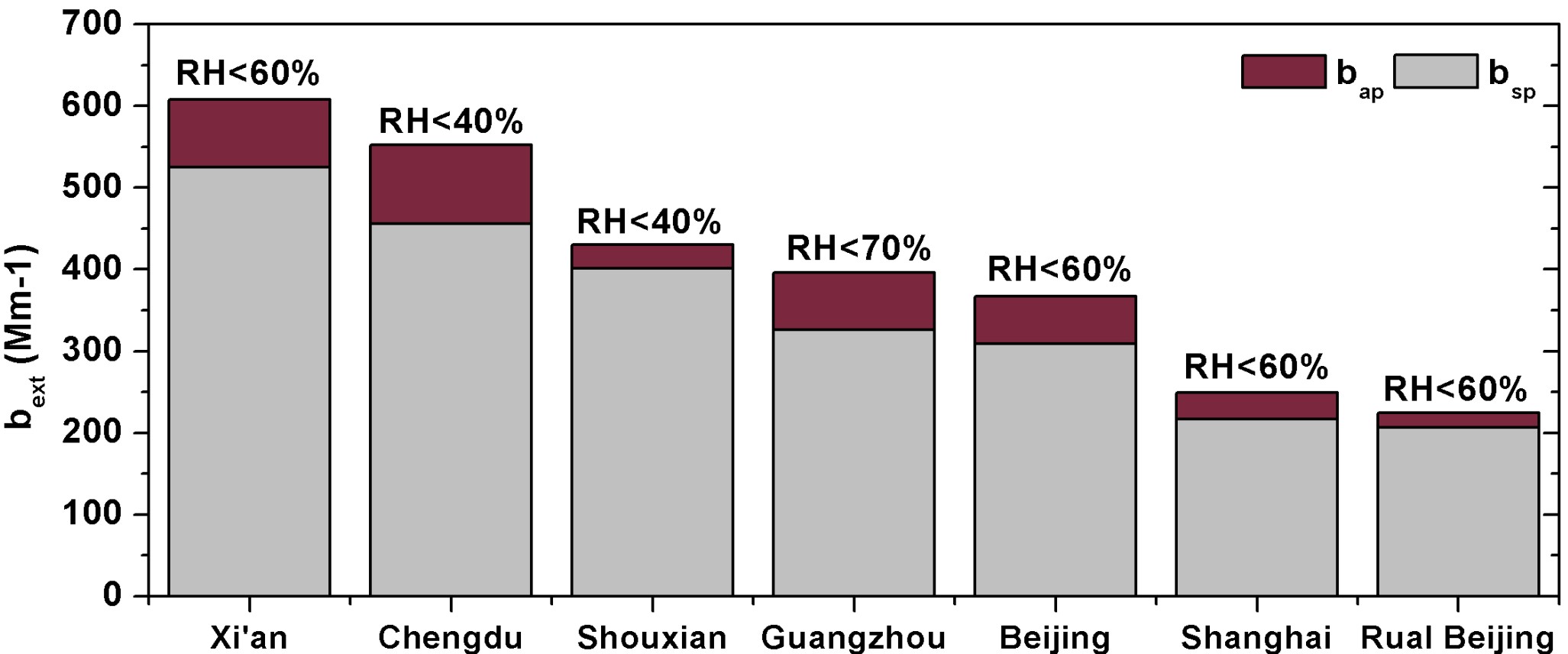

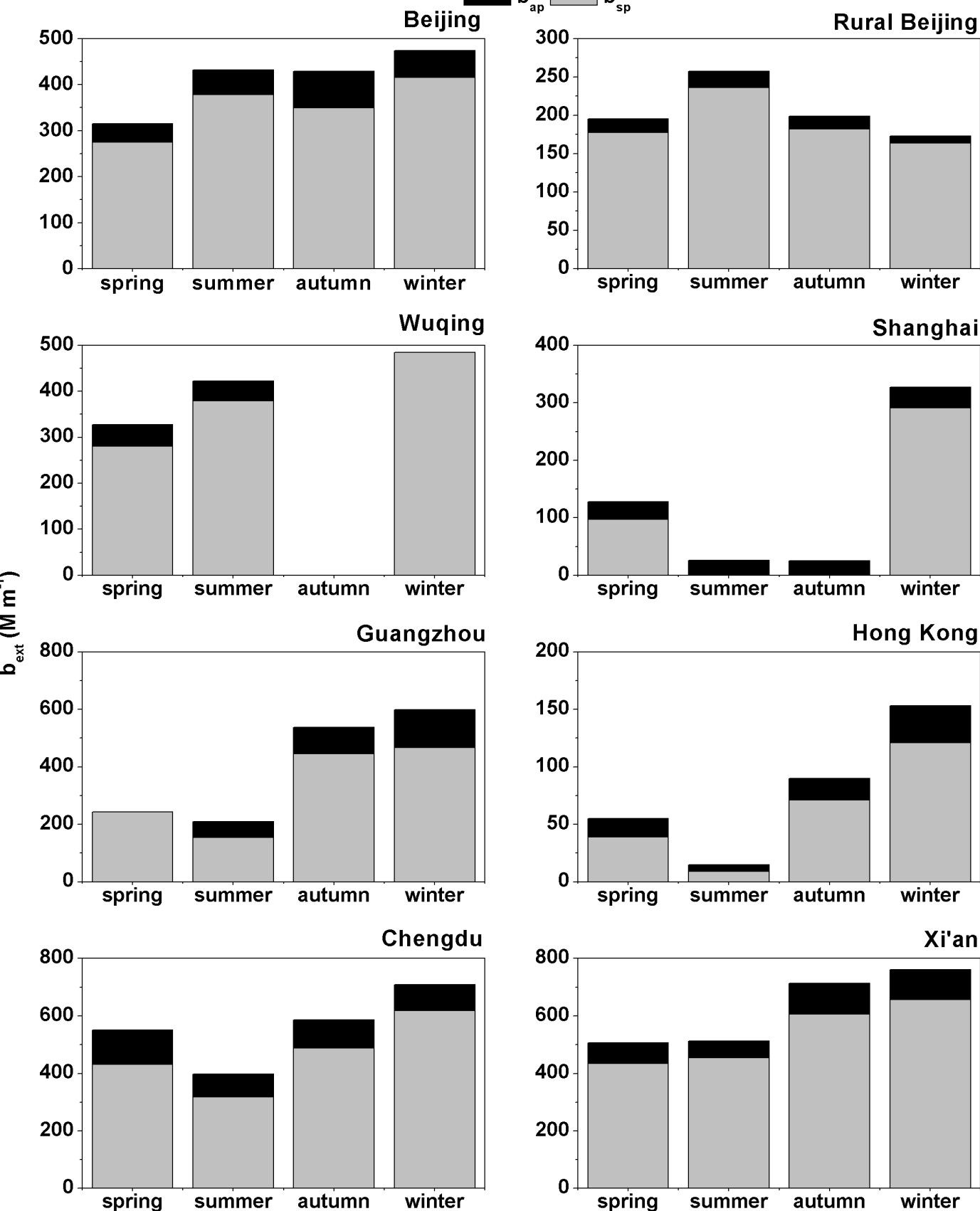

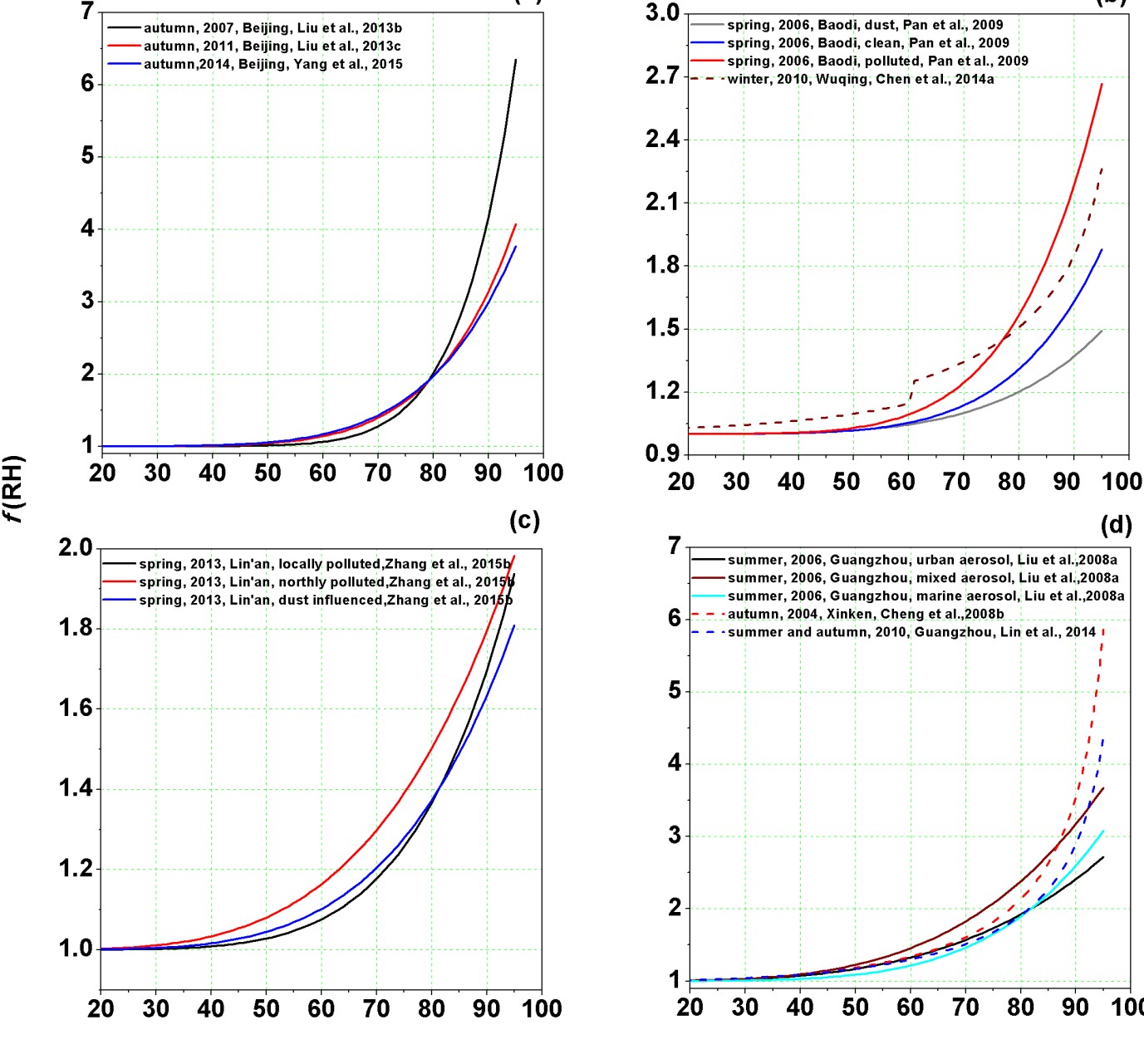

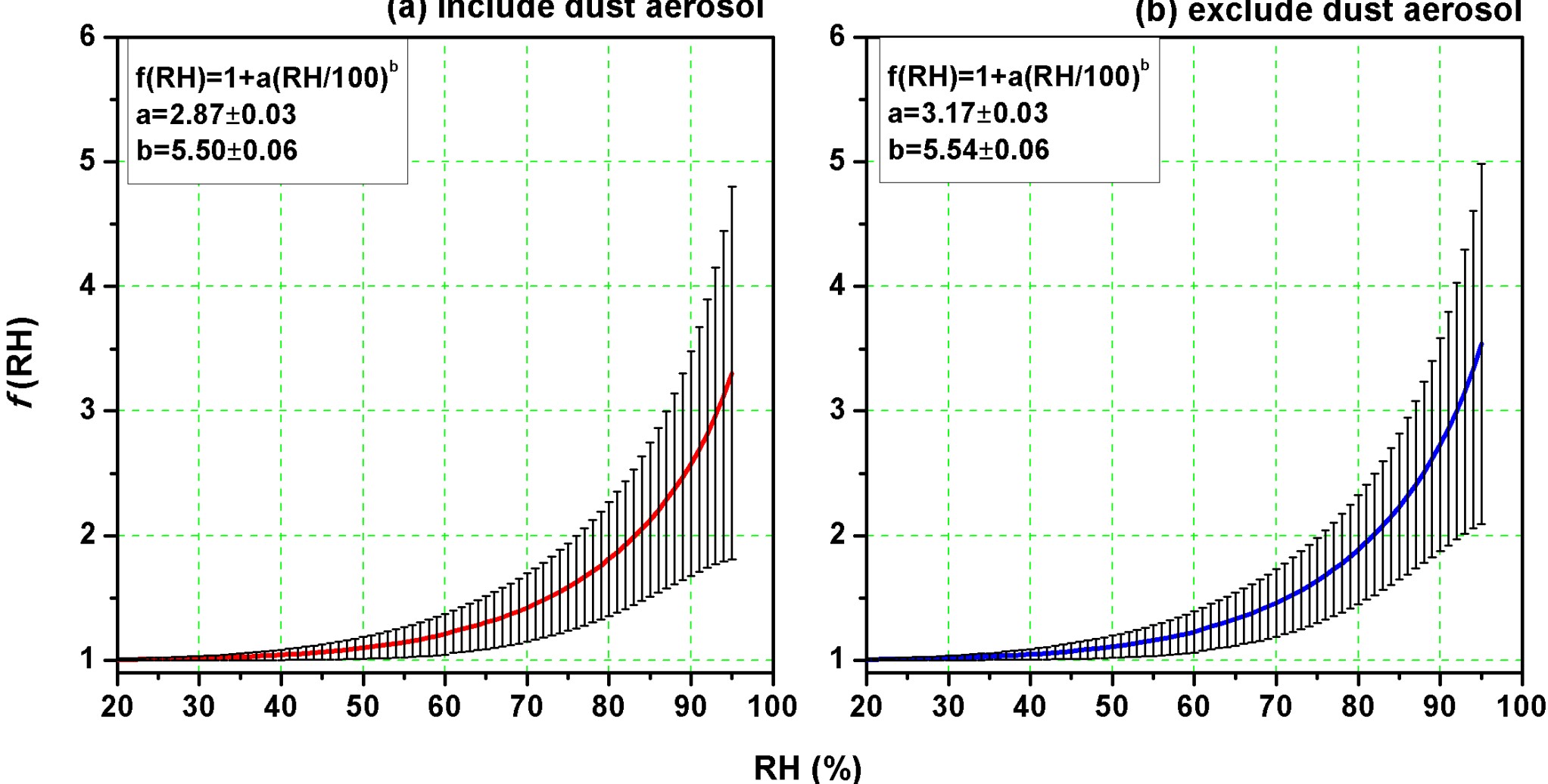

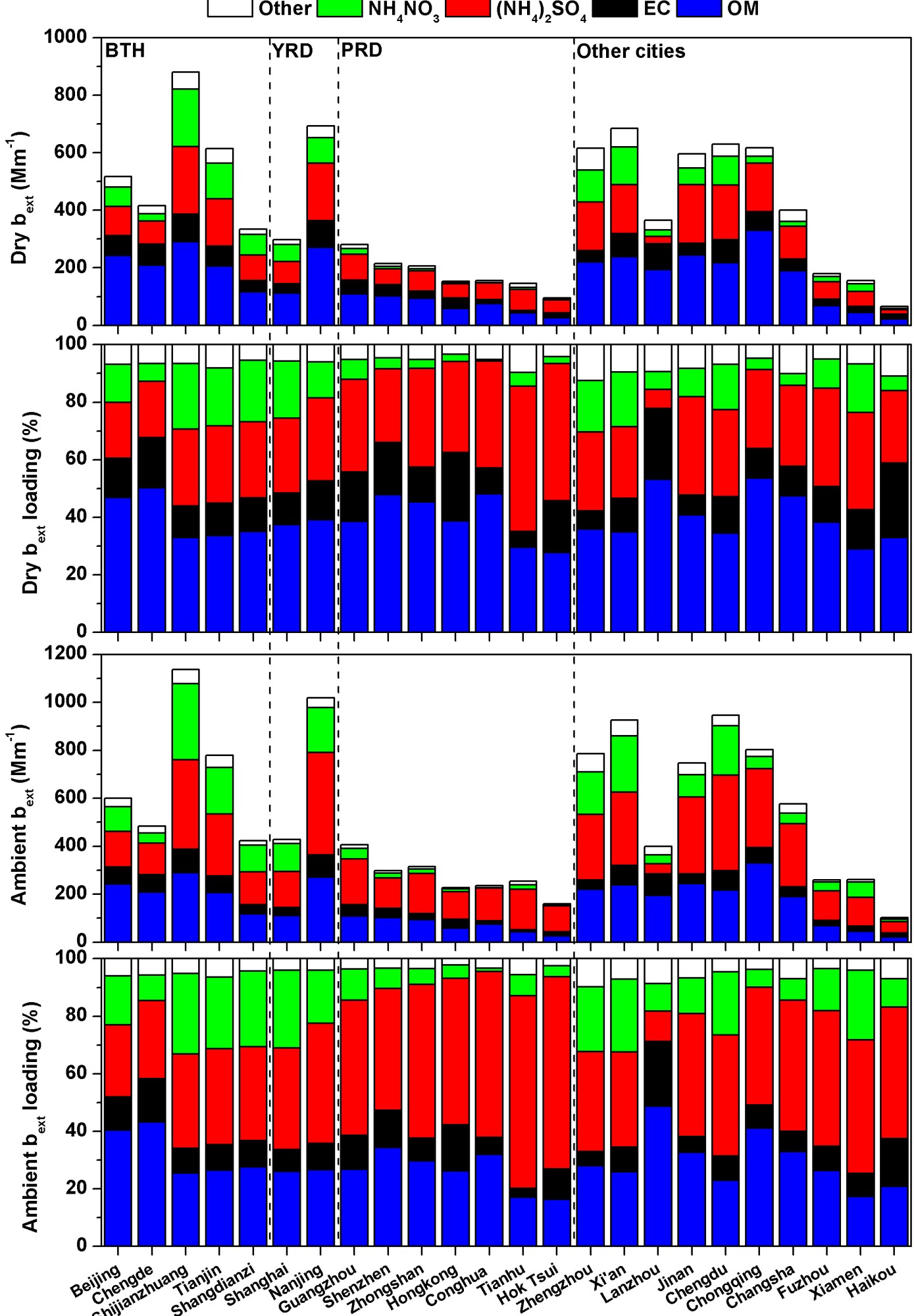