# Peer review of "A review of current knowledge concerning PM2.5 chemical composition, aerosol optical properties, and their relationships across China"

_Atmospheric Chemistry and Physics, 2017_

## Referee Comment (RC1) · Anonymous Referee #1 · 4 May 2017

This paper presents a literature survey on $PM_{2.5}$ chemical composition and aerosol optical properties conducted in China in the past nearly two decades. Geographical, inter-annual, and seasonal variations are discussed. Aerosol hygroscopic properties are investigated, source apportionments of haze formation are summarized, and historical emission policies are reviewed. A larger number of tables summarizing existing studies are also provided in the supplemental information document, providing a one-stop reference collection. The study provides the much needed knowledge that is useful for guiding future research needs and making emission control policies. The paper is well organized and well written. I have the following comments for the authors to consider in its final version.

1. An urgent task facing the Chinese government and the scientific community is to quantify the sources and formation mechanisms causing episodic events of high $PM_{2.5}$ mass concentrations and sever haze. This paper provides a summary of source appointment studies on haze events, but not on $PM_{2.5}$ mass concentrations. It is recommended to also include a review of source factors identified for $PM_{2.5}$ in various regions of China.

2. For a few cities such as Beijing, Shanghai and Guangzhou, inter-annual variations are discussed based on field measurements conducted by different researchers (and likely using different instruments and/or QA/QC methods). How much confidence do you have on these inter-annual variations compared to measurement uncertainties?

3. A related question to question 2 above: is it possible to compare the trends identified in this study to other sources such as the online $PM_{2.5}$ data, the AOD trend analysis data, or available literature?

4. Please also add sub-section titles in the content lists.

---

## Short Comment (SC1) · 4 May 2017

Dear authors, we think datasets from another campaign about aerosol hygroscopic properties should be included in this review paper. Reference: Deliquescent phenomena of ambient aerosols on the North China Plain, Geophys. Res. Lett., 8744-8750, 10.1002/2016GL070273, 2016

---

## Author Comment (AC1) · 4 May 2017

Thank you very much for pointing out this negligence. We have obtained a copy of this paper and are studying its results, and will include it in the revised version of our paper.

---

## Referee Comment (RC2) · Anonymous Referee #3 · 17 May 2017

General Comments Aerosol is very important to impact atmospheric cycle and climate system by direct and indirect effects, a hot issue of scientific researches internationally. The paper summarizes the recent published on Chinese PM2.5 and reviews the tempo-spatial distribution of PM2.5, chemical composition, aerosol optical properties, and reveals their relation across the whole country, based on ground-based filter measurements of particles, gases (e.g. SO2, NO2, CO). In fact, high aerosol burden regions such as areas in Asia are still not well characterized in terms of particle chemical and microphysical properties and long-term variation trend. The topic of this paper is of common interest within the scientific community. Although the manuscript includes some important data, however, the quality is not sufficient in the current state to be

directly published. The authors should take the suggestions made here into consideration for revision.

Specific suggestions 1. The paper mainly presents the PM2.5 measurements in urban sites, especially in eastern areas and other areas with relatively strong human activities (Figure 1). In addition, the variation of PM2.5 is very different in the North, the Middle and the South, so the authors should address it clearly (Figure 3). This paper somewhat provides more efforts to give a long-term trend of PM2.5. However, it lacks some remote sites such as in northeastern, Xinjiang, Yunnan areas etc, maybe it is better to select one typical year to focus on these sites and compare with the sites in the paper. 2. In lines of 620-624, AOD can reflect the column amount of aerosol in the whole atmosphere, while PM2.4 is only the mass of particles at the surface. The differences in fine structures of PM2.5 and AOD are related to PM2.5-AOD comparison and spatial variations of chemical composition, the size, number, vertical distribution and transport of aerosol are also responsible for these differences. The authors should address them clearly.

---

## Referee Comment (RC3) · Anonymous Referee #2 · 9 Jun 2017

The review paper jumps to the hot topics on PM2.5 pollution and aerosol optical property in China. It is well-written and very helpful for understanding the current situations and challenges ahead for alleviating severe PM2.5 pollution in China. This reviewer has a few minor comments for authors considering before publishing in ACP.

1) The authors are encouraged to use either OC or OM through the manuscript. 2) As a rural site downwind of BTH, the study by Feng et al. (JRG, 117, D03302, doi:10.1029/2011JD016400, 2012) is worthy of inclusion for comparison. 3) Lines 191-194, temperature effect should be included. A few very recent studies suggest that extreme weather could also be important factors for heavy PM2.5 pollution in winter.

[Figure]

The authors may have no time to read, but these studies are really worthy of inclusion for a complete review. 4) Lines 288-295, the reviewer suggest to include these contentious studies for sulfate formation in atmospheric particles published in 2016 and add a few arguments as well. It is helpful for students and young scientists. 5) Lines 616-624, relative humidity is also important factor to determine spatial variation of AOD. 6) In Section "4.3 Aerosol gyroscopic properties", the authors are encouraged to include aerosol particle size information if possible.

---

## Author Comment (AC2) · 27 Jun 2017

**Response to Referee #1**

We greatly appreciate the helpful comments from the reviewer, which have helped us improve the paper. We have addressed the comments carefully, as detailed below.

1. An urgent task facing the Chinese government and the scientific community is to quantify the sources and formation mechanisms causing episodic events of high $PM_{2.5}$ mass concentrations and sever haze. This paper provides a summary of source appointment studies on haze events, but not on $PM_{2.5}$ mass concentrations. It is recommended to also include a review of source factors identified for $PM_{2.5}$ in various regions of China.

Response: We have found more than 40 SCI articles on $PM_{2.5}$ source-appointment studies published during 2000-2017. We have provided a summary table (Table S2) of these studies in the SI document, and added a new section (2.3) in the revised paper focusing on source-apportionment studies. In this section, we first briefly summarized common receptor models used in $PM_{2.5}$ source-appointment studies and common source factors found in Chines cities. We then discussed annual and seasonal contributions of dominant source factors to $PM_{2.5}$ mass region by region.

In the abstract, we have also provided a summary of major findings based on the review of these studies, which reads: "Source apportionment analysis identified secondary inorganic aerosols, coal combustion, and traffic emission as the top three source factors contributing to $PM_{2.5}$ mass in most Chinese cities, and the sum of these three source factors explained 44% to 82% of $PM_{2.5}$ mass across China. Biomass emission in any cities, industrial emission in industrial cities, dust emission in northern cities, and ship emission in coastal cities are other major source factors, each of which contributed 7-27% to $PM_{2.5}$ mass in applicable cities.

2. For a few cities such as Beijing, Shanghai and Guangzhou, inter-annual variations are discussed based on field measurements conducted by different researchers (and likely using different instruments and/or QA/QC methods). How much confidence do you have on these inter-annual variations compared to measurement uncertainties?

Response: We have carefully collected the information about the measurement and analysis methods used in literature and identified potential measurement uncertainties for the dominant chemical components (OC/EC and water soluble inorganic ions). We have added this information in section 2.2 in the revised paper:

"To ensure the comparability of the data collected using different instruments, measurement uncertainties were first briefly discussed here. Most studies in China analyzed OC and EC using DRI carbon analyzer or Sunset carbon analyzer. IMPROVE is the most widely used thermal/optical protocol for OC and EC analysis for DRI analyzer while NIOSH is the one for Sunset analyzer. OC and EC measured

by the two analyzers are comparable if using the same analysis protocol. For example, Wu et al. (2011) showed that OC from Sunset analyzer was only 8% lower than that from DRI analyzer, while EC was only 5% higher. However, when using different protocols by the two analyzers, the differences were much larger, e.g., EC from NIOSH was almost 50% lower than that from IMPROVE (Chow et al., 2010; Yang et al., 2011a). Note that OC and EC were also measured using a CHN elemental analyzer in 2001-2002 in Beijing, which protocol was similar to NIOSH (Duan et al., 2006). In any case, the measurement uncertainties of total carbon (TC, the sum of OC and EC) were less than 10% (Chow et al., 2010; Wu et al., 2011).

The ions including $SO_4^{2-}$, $NO_3^-$ and $NH_4^+$ were measured by ion chromatograph. Measurement uncertainties should be less than 15% in most cases under strict QA/QC procedures (Orsini et al., 2003; Trebs et al., 2004; Weber et al., 2003), but could be larger for ammonium nitrate ($NH_4NO_3$) since it can evaporate from the filters before chemical analysis under high temperature and low relative humidity (RH) conditions, and this applies to both quartz fiber filter and Teflon filter (Keck and Wittmaack, 2005; Weber et al., 2003). The loss of $NO_3^-$ due to evaporation was found to range from 4% to 84% depending on ambient temperature (Chow et al., 2005). Although the exact magnitudes of measurement uncertainties cannot be determined for $NO_3^-$ and $NH_4^+$, they are expected not to affect significantly the inter-annual variations discussed below for the three cities (Beijing, Shanghai, and Guangzhou) considering the small year-to-year temperature changes."

We have taken into account the above information when discussing the trends of measured species through this section.

3. A related question to question 2 above: is it possible to compare the trends identified in this study to other sources such as the online $PM_{2.5}$ data, the AOD trend analysis data, or available literature?

Response: As noted in a recent paper by Fontes (2017): "The long trends of $PM_{2.5}$ concentrations were not fully investigated in China, in particular the year-to-year trends and the seasonal and daily cycles." They analyzed $PM_{2.5}$ data from 1999-2008 at five megacities in China. We have added this reference in the revised paper. The data set we collected in this review paper covered much longer periods and all the sites across China.

We have added a brief discussion on the relationship between AOD and $PM_{2.5}$ at the beginning of Section 3, which reads: "Satellite retrievals of AOD have been widely applied to estimate surface $PM_{2.5}$ concentrations using statistical models (Liu et al., 2005; Hu et al., 2013; Ma et al., 2014; Wang and Christopher, 2003). Although the correlation between AOD and $PM_{2.5}$ mass concentration depends on many factors, such as aerosol size distribution, refractive index, single-scattering albedo, and meteorological factors (Che et al., 2009; Guo et al., 2009b; Guo et al., 2017), the

predicted PM$_{2.5}$ mass from satellite AOD data compared well with ground-level measurements (Ma et al., 2014; Xie et al., 2015b). Moreover, the spatial distributions of AOD measured using sun photometers mostly agreed with those retrieved from satellite data (Che et al., 2014; Che et al., 2015; Liu et al., 2016b; Pan et al., 2010)."

4. Please also add sub-section titles in the content lists.

Response: We have added sub-section titles in the contents list.

---

## Author Comment (AC3) · 27 Jun 2017

**Response to Referee #3**

We greatly appreciate the helpful comments from the reviewer, which have helped us improve the paper. We have addressed the comments carefully, as detailed below.

General Comments
Aerosol is very important to impact atmospheric cycle and climate system by direct and indirect effects, a hot issue of scientific researches internationally. The paper summarizes the recent published on Chinese PM2.5 and reviews the tempo-spatial distribution of PM2.5, chemical composition, aerosol optical properties, and reveals their relation across the whole country, based on ground-based filter measurements of particles, gases (e.g. SO2, NO2, CO). In fact, high aerosol burden regions such as areas in Asia are still not well characterized in terms of particle chemical and microphysical properties and long-term variation trend. The topic of this paper is of common interest within the scientific community. Although the manuscript includes some important data, however, the quality is not sufficient in the current state to be directly published. The authors should take the suggestions made here into consideration for revision.

Specific suggestions
1. The paper mainly presents the $PM_{2.5}$ measurements in urban sites, especially in eastern areas and other areas with relatively strong human activities (Figure 1). In addition, the variation of $PM_{2.5}$ is very different in the North, the Middle and the South, so the authors should address it clearly (Figure 3). This paper somewhat provides more efforts to give a long-term trend of $PM_{2.5}$. However, it lacks some remote sites such as in northeastern, Xinjiang, Yunnan areas etc, maybe it is better to select one typical year to focus on these sites and compare with the sites in the paper.

Response: As explained at the beginning of section 2, the purpose of the study is to summarize chemically-resolved $PM_{2.5}$ data across China. Thus, only data sets have synchronous measurements of $PM_{2.5}$ and its major chemical components (inorganic ions, OC and EC) are included in this review. We are aware that they are many other studies concerning $PM_{2.5}$ pollution in many regions of China, however, most of these studies do not have information on $PM_{2.5}$ chemical composition and thus cannot be included in this review. We have gone through a more careful literature survey and found a few additional studies conducted in medium-sized cities and remote sites. These studies have been added in the SI tables and numbers in the paper due to the addition of these studies have been updated.

2. In lines of 620-624, AOD can reflect the column amount of aerosol in the whole atmosphere, while $PM_{2.5}$ is only the mass of particles at the surface. The differences in fine structures of $PM_{2.5}$ and AOD are related to $PM_{2.5}$-AOD comparison and spatial variations of chemical composition, the size, number, vertical distribution and transport of aerosol are also responsible for these differences. The authors should

address them clearly.

Response: We have added some materials and relevant references explaining the relationship between AOD and $PM_{2.5}$ in section 3, which reads: "Satellite retrievals of AOD have been widely applied to estimate surface $PM_{2.5}$ concentrations using statistical models (Liu et al., 2005; Hu et al., 2013; Ma et al., 2014; Wang and Christopher, 2003). Although the correlation between AOD and $PM_{2.5}$ mass concentration depends on many factors, such as aerosol size distribution, refractive index, single-scattering albedo, and meteorological factors (Che et al., 2009; Guo et al., 2009b; Guo et al., 2017), the predicted $PM_{2.5}$ mass from satellite AOD data compared well with ground-level measurements (Ma et al., 2014; Xie et al., 2015b). Moreover, the spatial distributions of AOD measured using sun photometers mostly agreed with those retrieved from satellite data (Che et al., 2014; Che et al., 2015; Liu et al., 2016b; Pan et al., 2010)."

---

## Author Comment (AC4) · 27 Jun 2017

**Response to Referee #2**

We greatly appreciate the helpful comments from the reviewer, which have helped us improve the paper. We have addressed the comments carefully, as detailed below.

The review paper jumps to the hot topics on PM2.5 pollution and aerosol optical property in China. It is well-written and very helpful for understanding the current situations and challenges ahead for alleviating severe PM2.5 pollution in China. This reviewer has a few minor comments for authors considering before publishing in ACP

1. The authors are encouraged to use either OC or OM through the manuscript.

Response: After a careful consideration, we feel both OC and OM are needed in the discussions in various places. For example, OC is measured directly and needed to be discussed in the measurement data as well as in comparing related emission inventories. OM is converted from OC using different conversion factors in different regions and is needed in assessing $PM_{2.5}$ mass distributions among different chemical components.

2. As a rural site downwind of BTH, the study by Feng et al. (JRG, 117, D03302, doi:10.1029/2011JD016400, 2012) is worthy of inclusion for comparison.

Response: We have included this reference and the following discussion in the revised paper: "At an Asian continental outflow site (Penglai in Shandong province), annual average contribution of secondary inorganic aerosols to $PM_{2.5}$ reached to 54% (Feng et al., 2012b), evidently higher than those in urban and inland rural sites in China, while that of carbonaceous aerosols was 31%, close to those in BTH. This finding suggested that intensive emissions of $SO_2$ and $NO_x$ in China enhanced the downward transport of secondary inorganic aerosols to Pacific Ocean."

3. Lines 191-194, temperature effect should be included. A few very recent studies suggest that extreme weather could also be important factors for heavy $PM_{2.5}$ pollution in winter. The authors may have no time to read, but these studies are really worthy of inclusion for a complete review.

Response: We have added the following discussion in the revised paper: "Moreover, extreme weather events such as weakening monsoon circulation, depression of strong cold air activities, strong temperature inversion, and descending air motions in the planetary boundary layer also played important roles in wintertime heavy $PM_{2.5}$ pollution (Niu et al., 2010; Wang et al., 2014c; Zhao et al., 2013). Several extreme wintertime air pollution events in recent years covered vast areas of northern China and were all correlated to some extent with extreme weather conditions (Zou et al., 2017)."

4. Lines 288-295, the reviewer suggest to include these contentious studies for sulfate formation in atmospheric particles published in 2016 and add a few arguments as well. It is helpful for students and young scientists.

Response: We have added the following discussion in the revised paper: "It is worth to note that several recent studies have highlighted the important role $NO_2$ might play in sulfate formation in the polluted environment in China (Cheng et al., 2016; Wang et al., 2016c; Xie et al., 2015a). Nevertheless, the aqueous $SO_2 + H_2O_2/O_3$ oxidation should still be the dominant mechanism in most cases, especially at a background site (Lin et al., 2017). The aqueous $SO_2$ + oxygen (catalyzed by Fe(III)) reaction can also be important under heavy haze condition in north China (Li et al., 2017b). Extensive measurements of stable oxygen are needed to confirm the relative contributions of different sulfate formation mechanisms."

5. Lines 616-624, relative humidity is also important factor to determine spatial variation of AOD.

Response: We have included meteorological factors (which cover RH) in AOD discussion in the revised paper, which reads: "Satellite retrievals of AOD have been widely applied to estimate surface $PM_{2.5}$ concentrations using statistical models (Liu et al., 2005; Hu et al., 2013; Ma et al., 2014; Wang and Christopher, 2003). Although the correlation between AOD and $PM_{2.5}$ mass concentration depends on many factors, such as aerosol size distribution, refractive index, single-scattering albedo, and meteorological factors (Che et al., 2009; Guo et al., 2009b; Guo et al., 2017), the predicted $PM_{2.5}$ mass from satellite AOD data compared well with ground-level measurements (Ma et al., 2014; Xie et al., 2015b). Moreover, the spatial distributions of AOD measured using sun photometers mostly agreed with those retrieved from satellite data (Che et al., 2014; Che et al., 2015; Liu et al., 2016b; Pan et al., 2010)."

6. In Section "4.3 Aerosol gyroscopic properties", the authors are encouraged to include aerosol particle size information if possible.

Response: We agree that particle size distribution is an important factor affecting $f$(RH) curves. However, size distributions of the dominant chemical components ($NH_4^+$, $SO_4^{2-}$, $NO_3^-$, OC, and EC) were only available in autumn of 2007 in urban Beijing. Thus, it is difficult to investigate the impact of different size distributions on the differences in $f$(RH) curves in different years (autumns of 2007, 2011 and 2014) in urban Beijing. For most of the other cities, only one study was available for $f$(RH) curves and particle size distribution data were also very limited. Thus, we chose not to go to the details on size-distribution related impacts. We, however, simply pointed out the potential influence of size distribution on $f$(RH) curves, which reads: "however, $f$(80%<RH<90%) values in rural Guangzhou were evidently higher than those in urban Guangzhou, likely due to the much higher fraction of secondary inorganic aerosols in fine mode particles in rural Guangzhou than urban Guangzhou in the dry

season (Lin et al., 2014; Liu et al., 2008b)."

Additional references added in the revised paper adding the comments from this reviewer are listed below:

Cheng, Y., Zheng, G., Wei, C., Mu, Q., Zheng, B., Wang, Z., Gao, M., Zhang, Q., He, K., Carmichael, G., Pöschl, U., and Su, H.: Reactive nitrogen chemistry in aerosol water as a source of sulfate during haze events in China, Science Advances, 2, 10.1126/sciadv.1601530, 2016.

Feng, J. L., Guo, Z. G., Zhang, T. R., Yao, X., Chan, C. K., and Fang, M.: Source and formation of secondary particulate matter in PM2.5 in Asian continental outflow, Journal of Geophysical Research, 117, 2012b.

Li, G., Bei, N., Cao, J., Huang, R., Wu, J., Feng, T., Wang, Y., Liu, S., Zhang, Q., Tie, X., and Molina, L. T.: A possible pathway for rapid growth of sulfate during haze days in China, Atmos. Chem. Phys., 17, 3301-3316, 10.5194/acp-17-3301-2017, 2017b.

Lin, M., Biglari, S., Zhang, Z., Crocker, D., Tao, J., Su, B., Liu, L., and Thiemens, M. H.: Vertically uniform formation pathways of tropospheric sulfate aerosols in East China detected from triple stable oxygen and radiogenic sulfur isotopes, Geophysical Research Letters, 44, doi:10.1002/2017GL073637, 2017.

Niu, F., Li, Z., Li, C., Lee, K., and Wang, M.: Increase of wintertime fog in China: Potential impacts of weakening of the Eastern Asian monsoon circulation and increasing aerosol loading, Journal of Geophysical Research, 115, 2010.

Wang, G., Zhang, R., Gomez, M. E., Yang, L., Zamora, M. L., Hu, M., Lin, Y., Peng, J., Guo, S., and Meng, J.: Persistent sulfate formation from London Fog to Chinese haze, Proceedings of the National Academy of Sciences, 113, 13630-13635, 2016c.

Wang, Y., Yao, L., Wang, L., Liu, Z., Ji, D., Tang, G., Zhang, J., Sun, Y., Hu, B., and Xin, J.: Mechanism for the formation of the January 2013 heavy haze pollution episode over central and eastern China, Science China Earth Sciences, 57, 14-25, 2014c.

Xie, Y., Ding, A., Nie, W., Mao, H., Qi, X., Huang, X., Xu, Z., Kerminen, V. M., Petaja, T., and Chi, X.: Enhanced sulfate formation by nitrogen dioxide: Implications from in situ observations at the SORPES station, Journal of Geophysical Research, 120, 12679-12694, 2015a.

Zhao, X. J., Zhao, P. S., Xu, J., Meng, W., Pu, W., Dong, F., He, D., and Shi, Q. F.: Analysis of a winter regional haze event and its formation mechanism in the North China Plain, Atmospheric Chemistry and Physics, 13, 5685-5696, 2013.

Zou, Y., Wang, Y., Zhang, Y., and Koo, J.H.: Arctic sea ice, Eurasia snow, and extreme winter haze in China, Science Advances, 3, 10.1126/sciadv.1602751, 2017.

---

## Author Response (AR2)

Dear Editor:

We have already addressed all of the comments provided by the three reviewers, as detailed in our responses to the reviewers' comments.

Thank you for taking care of the review process for this paper.

Sincerely,

Jun Tao, Leiming Zhang and co-authors